# Identification and dynamics of the human ZDHHC16-ZDHHC6 palmitoylation cascade

Laurence Abrami[1†], Tiziano Dallavilla[1,2†], Patrick A Sandoz[1], Mustafa Demir[1], Béatrice Kunz[1], Georgios Savoglidis[2], Vassily Hatzimanikatis[2*], F Gisou van der Goot[1*]

[1]Global Health Institute, Faculty of Life Sciences, Ecole Polytechnique Fédérale de Lausanne, Lausanne, Switzerland; [2]Laboratory of Computational Systems Biotechnology, Faculty of Basic Sciences, Ecole Polytechnique Fédérale de Lausanne, Lausanne, Switzerland

**\*For correspondence:**
vassily.hatzimanikatis@epfl.ch
(VH);
gisou.vandergoot@epfl.ch (FGG)

[†]These authors contributed
equally to this work

**Reviewing editor:** Philippe IH
Bastiaens, Max Planck Institute of
Molecular Physiology, Germany

**Abstract** S-Palmitoylation is the only reversible post-translational lipid modification. Knowledge about the DHHC palmitoyltransferase family is still limited. Here we show that human ZDHHC6, which modifies key proteins of the endoplasmic reticulum, is controlled by an upstream palmitoyltransferase, ZDHHC16, revealing the first palmitoylation cascade. The combination of site specific mutagenesis of the three ZDHHC6 palmitoylation sites, experimental determination of kinetic parameters and data-driven mathematical modelling allowed us to obtain detailed information on the eight differentially palmitoylated ZDHHC6 species. We found that species rapidly interconvert through the action of ZDHHC16 and the Acyl Protein Thioesterase APT2, that each species varies in terms of turnover rate and activity, altogether allowing the cell to robustly tune its ZDHHC6 activity.
DOI: https://doi.org/10.7554/eLife.27826.001

## Introduction

Cells constantly interact with and respond to their environment. This requires tight control of protein function in time and in space, which largely occurs through reversible post-translational modifications of proteins, such as phosphorylation, ubiquitination and S-palmitoylation. The latter consist in the addition on an acyl chain, generally C16 in mammals, to cysteine residues, thereby altering the hydrophobicity of the protein and tuning its function (*Blaskovic et al., 2014*; *Chamberlain and Shipston, 2015*). More precisely, palmitoylation may control the interaction of a protein with membranes or specific membrane domains, affect its conformation, trafficking, stability and/or activity (*Blaskovic et al., 2014*; *Chamberlain et al., 2013*; *Conibear and Davis, 2010*; *Fukata et al., 2016*). In the cytosol, the acyl chain is attached to the protein via a thioester bond through the action of DHHC palmitoyltransferases, a family of multispanning transmembrane proteins (*Blaskovic et al., 2014*; *Chamberlain et al., 2013*; *Conibear and Davis, 2010*; *Fukata et al., 2016*).

The list of proteins undergoing palmitoylation is ever increasing ([*Blanc et al., 2015*], http://swiss-palm.epfl.ch/) and the modification is found to be important in numerous key cellular processes including neuronal development and activity (*Fukata and Fukata, 2010*), cardiac function (*Pei et al., 2016*), systemic inflammation (*Beard et al., 2016*), innate immunity to viruses (*Mukai et al., 2016*), cell polarity (*Chen et al., 2016*), EGF-signalling (*Runkle et al., 2016*), protease activity (*Skotte et al., 2017*) and cancer (*Coleman et al., 2016*; *Thuma et al., 2016*).

While novel roles and targets of palmitoylation are constantly reported, little is known about the regulation and dynamics of this post-translational modification. Here we focused on one of the 23

human palmitoyltransferase, ZDHHC6, which localizes to the endoplasmic reticulum (ER) and controls a panel of key ER substrates such as the ER chaperone calnexin (*Lakkaraju et al., 2012*), the ER E3 ligase gp78 (*Fairbank et al., 2012*), the IP$_3$ receptor (*Fredericks et al., 2014*) as well as cell surface proteins such as the transferrin receptor (*Senyilmaz et al., 2015*). For each of these proteins, palmitoylation controls stability, localization, trafficking and/or function.

As most DHHC enzymes, ZDHHC6 is a tetra-spanning membrane protein. It has a short N-terminal extension and a long C-terminal tail composed of an approximately 100-residue domain of unknown structure followed by a variant SH3_2 domain (*Figure 1A*, *Figure 1—figure supplement 1*). At the C-terminus, it contains a KKNR motif which when transferred to the ZDHHC3 enzyme leads to its relocalization from the Golgi to the ER (*Gorleku et al., 2011*), suggesting that it contributes to the ER localization of ZDHHC6.

Here we show that ZDHHC6 function, localization and stability are all regulated through the dynamic multi-site palmitoylation of its SH3_2 domain. Interestingly, palmitoylation occurs via an upstream palmitoyltransferase, ZDHHC16, revealing for the first time the existence of palmitoylation cascades. Depalmitoylation is mediated by the Acyl Protein Thioesterase APT2. Palmitoylation can occur on three different sites, leading to potentially eight different ZDHHC6 species defined by acyl site occupancy. These species can interconvert through the actions of ZDHHC16 and APT2. Using mathematical modelling combined with various kinetic measurements on WT and mutant ZDHHC6, we probed the complexity of this system and its importance for ZDHHC6 activity. Altogether, this study shows that palmitoylation affects the quaternary assembly of ZDHHC6, its localization, stability and function. Moreover we show that the presence of 3 sites is necessary for the robust control of ZDHHC6 abundance upon changes in ZDHHC16 activity.

## Results

### Palmitoylation of the ZDHHC6 SH3_2 domain

*ZDHHC6* KO cells were generated using the CRISPR-cas9 system in the near haploid cell line HAP1. Using the Acyl-RAC method to isolate palmitoylated proteins (*Werno and Chamberlain, 2015*), we verified that the ER chaperone calnexin (*Lakkaraju et al., 2012*), the E3 ligase gp78 (*Fairbank et al., 2012*), the IP$_3$ receptor (*Fredericks et al., 2014*) and the transferrin receptor (Trf-R) (*Senyilmaz et al., 2015*) are indeed ZDHHC6 targets (*Figure 1B*). Interestingly calnexin and Trf-R were no longer captured by Acyl-RAC in the HAP1 ZDHHC6 KO cells, confirming that they are exclusively modified by ZDHHC6, while capture of the IP$_3$ receptor and gp78 was reduced but not abolished indicating that they can also be modified by other palmitoyltransferases (*Figure 1B*) consistent with previous findings (*Fairbank et al., 2012*; *Fredericks et al., 2014*). As negative controls, we tested flotillin1, a known target of ZDHHC5 (*Li et al., 2012*) and the anthrax toxin receptor one modified by a yet to be determined DHHC enzymes (*Abrami et al., 2006*).

When probing samples for ZDHHC6, we found that the palmitoyltransferase itself was captured by Acyl-RAC in HAP1 cells (*Figure 1B*) and also in different mouse tissues (shown for mouse lung tissue in *Figure 1C*). Palmitoylation of ZDHHC6 was reported in two large-scale proteomics analysis to occur on Cys-328, Cys-329 and Cys-343 in the SH3 domain (*Collins et al., 2017*; *Yang et al., 2010*). These cysteines are conserved in vertebrates (*Figure 1—figure supplement 1*). We generated single (ACC, CAC, CCA), double (AAC, CAA, ACA) and triple (AAA) cysteine-to-alanine mutants and monitored $^3$H-palmitate incorporation during 2 hr. Palmitoylation of ZDHHC6 WT, but not the AAA mutant, was readily detected (*Figure 1DE*). All single cysteine mutants, especially C328A, and all double mutants showed a decrease in the $^3$H-palmitate signal (*Figure 1DE*). Thus ZDHHC6 can indeed undergo palmitoylation, and importantly it has no palmitoylation sites other than the three in the SH3 domain.

### The ZDHHC16-ZDHHC6 palmitoylation cascade

We next investigated whether ZDHHC6 was palmitoylating itself, in *cis* or *trans*. We generated an inactive DHHC deletion mutant (ΔDHHC) as well as a cell line stably expressing an shRNA against the *ZDHHC6* 3'UTR (*Figure 2—figure supplement 1A*). WT and (ΔDHHC) ZDHHC6 both underwent palmitoylation when transiently expressed in control or shRNA *ZDHHC6* cells (*Figure 2—figure supplement 1A*), indicating that ZDHHC6 must be modified by another enzyme. We next performed

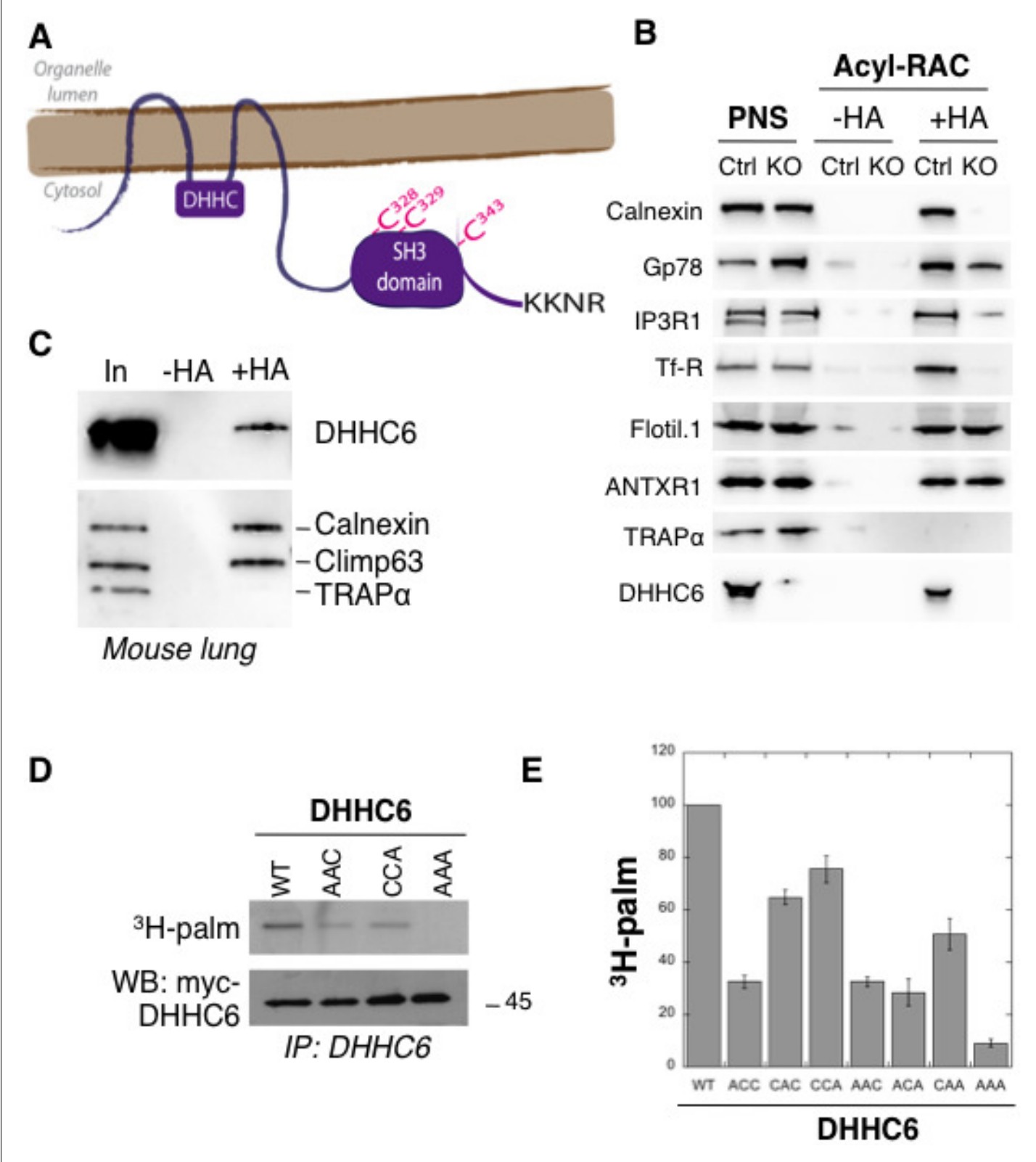

**Figure 1.** ZDHHC6 can undergo palmitoylation on three cysteines of its SH3 domain. (**A**) Schematic representation of ZDHHC6 enzyme. The potential ER retention motif is KKNR. (**B**) Analysis of protein acylation in control HAP cells (Ctrl) versus *ZDHHC6* KO HAP (KO). HAP cell membranes were recovered by centrifugation and incubated with MMTS and then with hydroxylamine (+HA) or with Tris (-HA) together with free thiol group binding beads. Eluted fractions were analysed by immunoblotting with the indicated antibodies. PNS represents 1/10 of the input fraction. (**C**) ZDHHC6

*Figure 1 continued on next page*

*Figure 1 continued*

acylation in mouse tissues. 400 μg total proteins extracted from mouse lung were incubated with MMTS and then with hydroxylamine (+HA) or with Tris (-HA) together with free thiol group binding beads. Eluted fractions were analysed by immunoblotting with the indicated antibodies. 'In' represents 1/10 of the input fraction. Calnexin and the ER shaping protein CLIMP-63 (*Lakkaraju et al., 2012*; *Schweizer et al., 1993*) were used as positive controls and the cysteine-less protein Trapα as negative control. (D) Palmitoylation of ZDHHC6 cysteine mutants. HeLa cells were transfected with plasmids encoding WT or the indicated Myc-tagged ZDHHC6 mutant constructs for 24 hr. Cells were then metabolically labelled for 2 hr at 37°C with $^3$H-palmitic acid. Proteins were extracted, immunoprecipitated with Myc antibodies, subjected to SDS-PAGE and analysed by autoradiography ($^3$H-palm), quantified using the Typhoon Imager or by immunoblotting with the indicated antibodies. (E) Quantification of $^3$H-palmitic acid incorporation into ZDHHC6. Quantified values were normalized to protein expression level. The calculated value of $^3$H-palmitic acid incorporation into WT ZDHHC6 was set to 100% and all mutants were expressed relative to this (n = 4, error bars represent standard deviation).

DOI: https://doi.org/10.7554/eLife.27826.002

The following figure supplement is available for figure 1:

**Figure supplement 1.** Alignment of DHHC6 sequences from different species.

DOI: https://doi.org/10.7554/eLife.27826.003

$^3$H-palmitate labelling experiments upon siRNA silencing of each of the 23 DHHC enzymes. While a significant but mild drop was observed for several DHHC enzymes, only silencing of *ZDHHC16* led to a major, 60%, drop in signal intensity (*Figure 2A*). $^3$H-palmitate incorporation was also reduced for the AAC and CCA mutants (*Figure 2—figure supplement 1B*), showing that all three sites are modified by the same enzyme.

We also performed a screen by over-expression of DHHC enzymes, which also pointed to ZDHHC16 as the responsible enzyme (*Figure 2B*, *Figure 2—figure supplement 1C*). The final confirmation was obtained using HAP1 cells in which specific DHHC enzymes were knocked out using the CRISPR-cas9 technology (*Figure 2C*). We focused on ZDHHCs 1, 7, 9, 12, 16 and 18, which led to some decrease in the siRNA screen (*Figure 2A*). ZDHHC6 palmitoylation was lost exclusively in *ZDHHC16* KO cells (*Figure 2C*). Consistent with these findings, co-immunoprecipitation experiments following transient over-expression of myc-tagged ZDHHC6 and FLAG-tagged ZDHHC16 showed that the two enzymes can interact (*Figure 2—figure supplement 1D*). Interestingly, *ZDHHC6* silencing in HeLa cells or *ZDHHC6* knock out in HAP1 cells both led to an increase in the *ZDHHC16* mRNA (*Figure 2—figure supplement 2AB*), indicating that these two proteins interact physically and genetically.

Altogether these experiments show that ZDHHC6 can be palmitoylated on all three of its SH3_2 cysteine residues by ZDHHC16, revealing for the first time that palmitoylation can occur in a cascade, as occurs for phosphorylation for example in the MAK kinase pathway. ZDHHC16, which localizes under over-expression conditions to the ER and the Golgi (*Figure 2—figure supplement 2A*), is not palmitoylated itself in HeLa cells, as shown both by $^3$H-palmitate incorporation and Acyl-RAC (*Figure 2—figure supplement 2BC*).

## Rapid APT2-mediated ZDHHC6 depalmitoylation

Palmitoylation is a reversible modification and thus has the potential to be dynamic. To analyse palmitate turnover on ZDHHC6, we performed $^3$H-palmitate pulse-chase experiments. Following a 2 hr pulse, 50% of the $^3$H-palmitate was lost from ZDHHC6 in ≈ 1 hr, indicating rapid turnover of the acyl chains (*Figure 2D*). We performed similar experiments on the various single and double cysteine mutants and found that rapid turnover required the presence of Cys-328. In its absence, the apparent palmitate release rates were considerably slower, reaching more than 4 hr (*Figure 2D*).

Depalmitoylation is an enzymatic reaction that is mediated by poorly characterized Acyl Protein Thioesterases (APT). We tested the involvement of APT1 and APT2 (*Blaskovic et al., 2014*), which are themselves palmitoylated on Cys-2 (*Kong et al., 2013*). We generated palmitoylation deficient variants of APT1 and 2 as well as catalytically inactive versions (APT1 S119A and APT2 S121A). Over-expression of WT or mutant APT1 had no detectable effect on ZDHHC6 palmitoylation (*Figure 2E, F*). In contrast, overexpression of WT, but not palmitoylation deficient, APT2 led to a significant decrease in ZDHHC6 palmitoylation. APT2$^{S121A}$ had an intermediate effect, possibly due to the formation of heterodimers between mutant and endogenous APT2 (*Kong et al., 2013*; *Vujic et al., 2016*). These observations show that ZDHHC6 palmitoylation is dynamic, in particular on Cys-328, and that depalmitoylation is mediated by APT2.

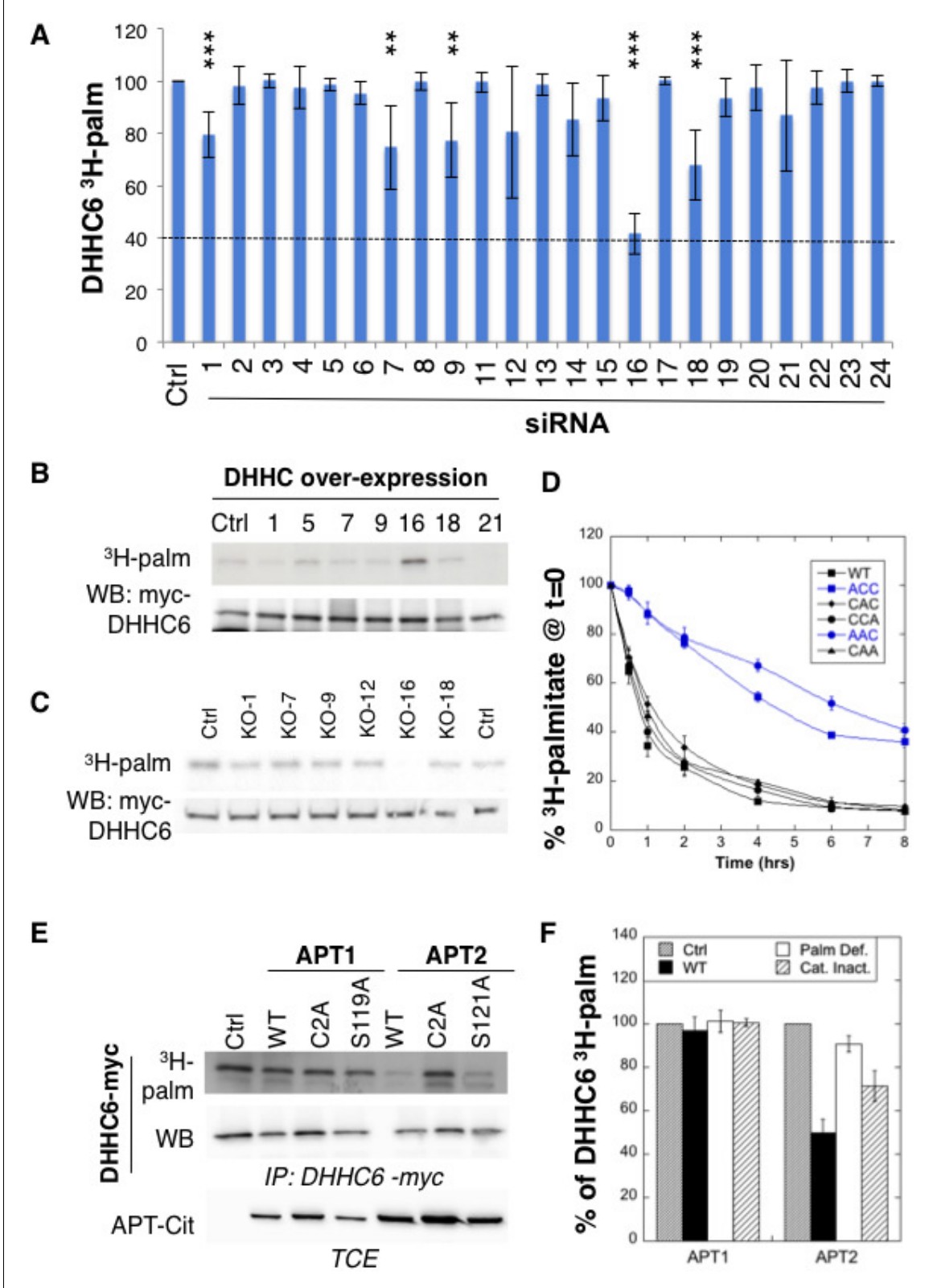

**Figure 2.** ZDHHC6 is palmitoylated by ZDHHC16 and depalmitoylated by APT2. (**A**) Identification of the ZDHHC6 palmitoyltransferase by siRNA screening of DHHC enzymes. HeLa cells were transfected with siRNAs against the indicated ZDHHC enzyme for 72 hr and with the myc-tagged WT ZDHHC6 expressing construct for the last 24 hr. Cells were then metabolically labelled 2 hr at 37°C with $^3$H-palmitic acid. Proteins were extracted, immunoprecipitated with myc antibodies and subjected to SDS-PAGE and analysed by autoradiography, quantified using the Typhoon Imager or by

*Figure 2 continued on next page*

*Figure 2 continued*

immunoblotting with myc antibodies. $^3$H-palmitic acid incorporation into ZDHHC6 was quantified and normalized to protein expression levels. The calculated value of $^3$H-palmitic acid incorporation into ZDHHC6 was set to 100% for an irrelevant siRNA (Ctrl) and all siRNA were expressed relative to this (n = 6, error bars represent standard deviation). (B) Identification of the ZDHHC6 palmitoltransferase by DHHC over-expression. HeLa cells were transfected with indicated the ZDHHC constructs and with myc-tagged WT ZDHHC6 construct for 24 hr. Cells were then metabolically labelled 2 hr at 37°C with $^3$H-palmitic acid. Proteins were extracted, immunoprecipated with myc antibodies and subjected to SDS-PAGE and analysed by autoradiography ($^3$H-palm) or by immunoblotting with myc antibodies. (C) Analysis of ZDHHC6 acylation in control HAP cells (Ctrl) versus HAP cells KO for DHHC 1, 7, 9, 12, 16, 18 (KO-X). Cells were transfected with the myc-tagged WT ZDHHC6 construct for 24 hr, then metabolically labelled 2 hr at 37°C with $^3$H-palmitic acid. Proteins were extracted, immunoprecipated with myc antibodies and subjected to SDS-PAGE and analysed by autoradiography ($^3$H-palm) or by immunoblotting with myc antibodies. (D) Palmitoylation decay of WT or mutant ZDHHC6. HeLa cells were transfected with plasmids encoding WT or the indicated mutant Myc-tagged ZDHHC6 constructs for 24 hr. Cells were then metabolically labelled 2 hr at 37°C with $^3$H-palmitic acid, washed and incubated with complete medium for different hours. Proteins were extracted, immunoprecipated with myc antibodies and subjected to SDS-PAGE and analysed by autoradiography, quantified using the Typhoon Imager or by immunoblotting with anti-myc antibodies. $^3$H-palmitic acid incorporation was quantified for each time point, normalized to protein expression level. $^3$H-palmitic acid incorporation was set to 100% at t = 0 after the 2 hr pulse and all different times of chase were expressed relative to this (n = 3, error bars represent standard deviation). (E) ZDHHC6 palmitoylation upon APT overexpression. HeLa cells were transfected with plasmids encoding myc-tagged WT ZDHHC6 and the indicated mutant citrine-tagged APT1 or APT2 constructs for 24 hr. APT2 was always expressed at a higher level that APT1. Cells were then metabolically labelled 2 hr at 37°C with $^3$H-palmitic acid. Proteins were extracted, immunoprecipated with myc antibodies and subjected to SDS-PAGE and analysed by autoradiography ($^3$H-palm), quantified using the Typhoon Imager or by immunoblotting with anti-myc antibodies. (F) Quantification of $^3$H-palmitic acid incorporation into ZDHHC6. Quantified values were normalized to protein expression level. $^3$H-palmitic acid incorporation was set to 100% for control cells (Ctrl) and values obtained for APT overexpressing cells were expressed relative to this (n = 6, error bars represent standard deviation).
DOI: https://doi.org/10.7554/eLife.27826.004

The following figure supplements are available for figure 2:

**Figure supplement 1.** ZDHHC16 mediates ZDHHC6 palmitoylation.
DOI: https://doi.org/10.7554/eLife.27826.005
**Figure supplement 2.** ZDHHC16 localizes to the early secretory pathway and is not palmitoylated.
DOI: https://doi.org/10.7554/eLife.27826.006
**Figure supplement 3.** Expression levels of ZDHHC6 and 16 and their influence on one another.
DOI: https://doi.org/10.7554/eLife.27826.007

## Palmitoylation of Cys-328 accelerates turnover of ZDHHC6

We next analysed the effect of palmitoylation on ZDHHC6 turnover. Palmitoylation has been reported to affect protein stability, increasing the half-life of proteins such as calnexin (*Dallavilla et al., 2016*) and the death receptor Fas (*Rossin et al., 2015*) but decreasing that of others such as gp78 (*Fairbank et al., 2012*). Protein turnover was monitored using $^{35}$S-Cys/Met metabolic pulse-chase. HeLa cells, transiently expressing myc-ZDHHC6, were submitted to a 20 min metabolic pulse followed by different times of chase before lysis and anti-myc immunoprecipitation, SDS-PAGE and auto-radiography. Decay of newly synthesized ZDHHC6 was biphasic, with 40% undergoing gradual degradation during the first 5 hr, and the remaining 60% undergoing degradation at a greatly reduced rate (*Figure 3A*). The overall apparent half-life was $t_{1/2}^{app} \approx 16$ hr. To enhance ZDHHC6 palmitoylation, we overexpressed ZDHHC16 or silenced *LYPLA2* (gene encoding APT2) expression. Both these genetic manipulations led to a dramatic acceleration of ZDHHC6 decay, with $t_{1/2}^{app} \approx$ 2 hr and 3 hr upon ZDHHC16 over expression and *Lypla2* silencing, respectively (*Figure 3A*). Silencing of *LYPLA1* (APT1) in contrast had no effect (*Figure 3—figure supplement 1A*). Remarkably, mutation of Cys-328, but not of Cys-329 or Cys-343, abolished the sensitivity to ZDHHC16 overexpression or *LYPLA2* silencing (*Figure 3BC*, *Figure 1—figure supplement 1BC*). *LYPLA2* silencing led to ubiquitination of ZDHHC6 (*Figure 3D*) and ZDHHC6 degradation could be rescued by the proteasome inhibitor MG132 (*Figure 3E*). Thus altogether these observations indicate that palmitoylation of Cys-328 renders ZDHHC6 susceptible to degradation by the ERAD pathway.

## Palmitoylation-dependent ZDHHC6 localization

We next investigated whether palmitoylation affects ZDHHC6 localization. Ectopically expressed WT myc-ZDHHC6 shows a typical ER staining, co-localizing with Bip, a lumenal ER chaperone, and BAP31, a transmembrane ER protein (*Figure 4AB*), as did all the cysteine mutants (*Figure 4—figure*

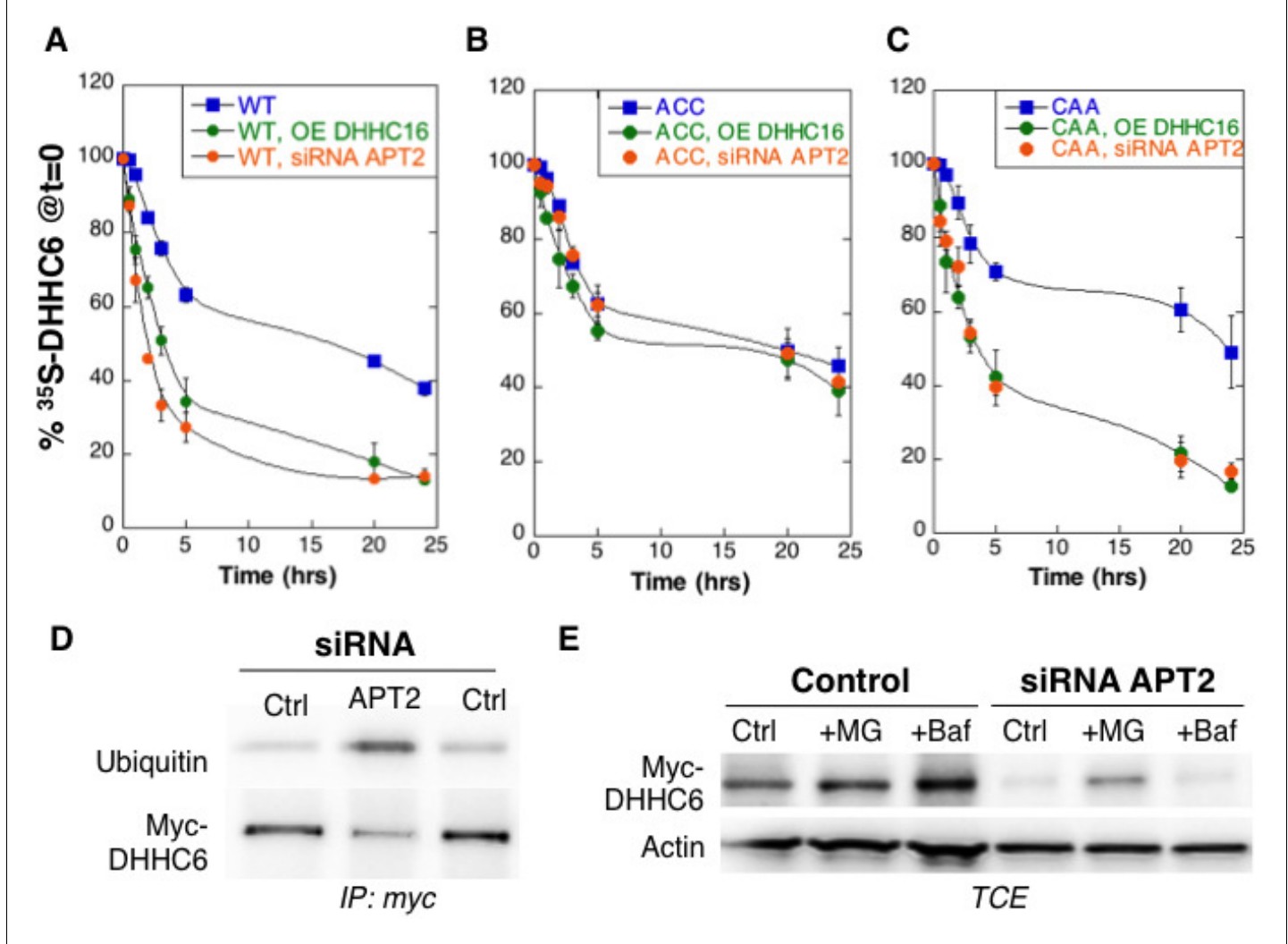

**Figure 3.** Palmitoylation on Cys-328 targets ZDHHC6 to ERAD. (A–C) Degradation kinetics of ZDHHC6. HeLa cells were transfected with plasmids encoding myc-tagged WT ZDHHC6 or cysteine mutants with or without FLAG-tagged WT ZDHHC16 for 24 hr after 48 hr transfection with siRNA *LYPLA2* (gene encoding APT2) or with control siRNA. HeLa cells were incubated for 20 min with $^{35}$S-Met/Cys at 37°C, washed and further incubated for different times at 37°C in complete medium. ZDHHC6 was immunoprecipitated and subjected to SDS-PAGE and analysed by autoradiography, quantified using the Typhoon Imager, and western blotting with anti-myc antibodies. $^{35}$S-Met/Cys incorporation was quantified for each time point, normalized to protein expression levels. $^{35}$S-Met/Cys incorporation was set to 100% for t = 0 after the 20 min pulse and all different times of chase were expressed relative to this (n = 3, error bars represent standard deviation). (D,E) ZDHHC6 ubiquitination and proteasomal degradation. HeLa cells were transfected with plasmids encoding myc-tagged WT ZDHHC6 constructs for 24 hr after 48 hr transfection with control (Ctrl) or *LYPLA2* (gene encoding APT2) siRNA. Proteins were extracted; ZDHHC6 was immunoprecipitated, subjected to SDS-PAGE and then analysed by immunoblotting with anti-ubiquitin or anti-myc antibodies (D). (E) Cells were treated 4 hr with 10 μM MG132 or with 100 nM Bafilomycin A before proteins were extracted; 40 μg of total extract were subjected to SDS-PAGE and then analysed by immunoblotting against actin, used as equal loading control, or myc.
DOI: https://doi.org/10.7554/eLife.27826.008

The following figure supplement is available for figure 3:

**Figure supplement 1.** Effect of palmitoylation on ZDHHC& decay.
DOI: https://doi.org/10.7554/eLife.27826.009

*supplement 1*). Ectopically expressed WT ZDHHC6 sometimes also localized to a dot (*Figure 4AB*). The ZDHHC6 AAA mutant was also clearly present in the reticular ER, but it tended to accumulate more frequently in dot-like ER structures, which stained positive for BAP31 but not Bip (*Figure 4AB*). Thus the inability of ZDHHC6 AAA to undergo palmitoylation influences its localization within the two dimensional space of the ER membrane.

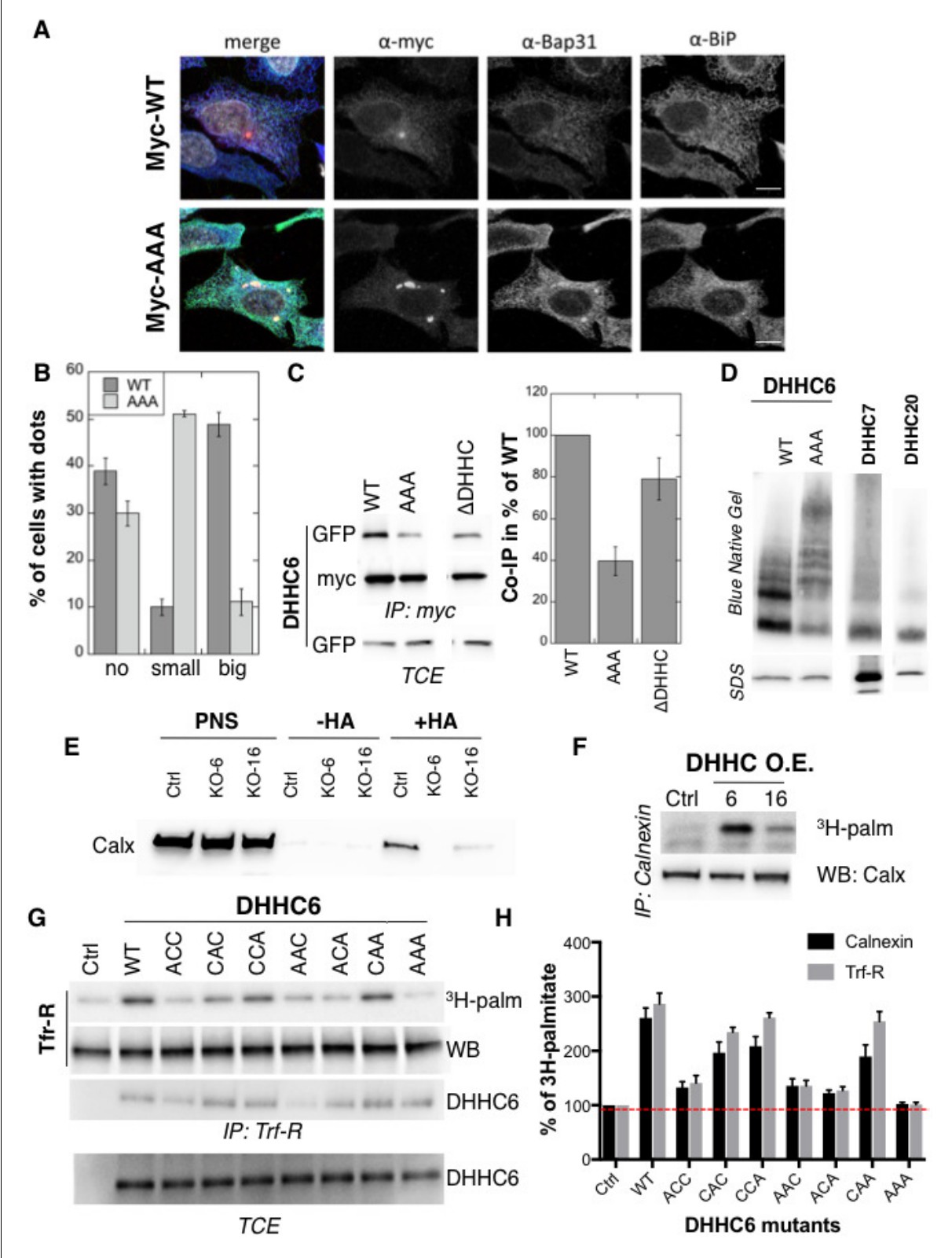

**Figure 4.** Palmitoylation affects assembly, localisation and function of ZDHHC6. (A,B) Immunofluorescence staining of HeLa cells transfected with myc-ZDHHC6 WT or AAA (bar = 10 μm). The presence of dots (small or big) was reported for each cell and quantified (data from three independent experiments, dots in 30 cells per condition, per experiment were counted, errors represent standard deviations). (C) Co-immunoprecipitation of ZDHHC6 variants. HeLa cells were transfected with plasmids encoding WT myc-ZDHHC6 and the indicated GFP-tagged mutant for 24 hr. Proteins were

*Figure 4 continued on next page*

*Figure 4 continued*

extracted, a total cell extract was analysed (TCE) and proteins were immunoprecipitated with myc antibodies and subjected to SDS-PAGE, then analysed by immunoblotting with anti-myc or anti-GFP antibodies. Quantification was performed by densitometry. The calculated value of co-immunoprecipitation with WT ZDHHC6 was set to 100%, all ZDHHC6 mutants were expressed relative to this (n = 4, errors represent standard deviations). (D) ZDHHC6 complexes. HeLa cells were transfected with plasmids encoding myc-ZDHHC6 WT or mutant, or myc-tagged DHHC7 or DHHC20 for 24 hr. Proteins were extracted, 40 μg of a TCE was analysed by SDS-PAGE or on blue native gels then analysed by immunoblotting with anti-myc antibodies. (E) Analysis of endogenous calnexin acylation in control HAP cells (Ctrl) versus HAP cells KO for *ZDHHC6* (KO-6) or for *ZDHHC16* (KO-16). HAP cell membranes were recovered by centrifugation and incubated with MMTS and then with hydroxylamine (+HA) or with Tris (−HA) together with free thiol group binding beads. Eluted fractions were analysed by immunoblotting with anti-calnexin antibodies. PNS represents 1/10 of the input fraction. (F) Analysis of endogenous calnexin palmitoylation in HeLa cells overexpressing ZDHHC6 or ZDHHC16. HeLa cells were transfected with plasmids encoding WT myc-ZDHHC6 or WT myc-ZDHHC16 constructs for 24 hr. Cells were then metabolically labelled 2 hr at 37°C with $^3$H-palmitic acid. Proteins were extracted and immunoprecipitated with anti-calnexin antibodies, subjected to SDS-PAGE and analysed by autoradiography ($^3$H-palm) or by immunoblotting with anti-calnexin antibodies. (G,H) Analysis of endogenous calnexin and Transferin receptor (Trf-R) palmitoylation in HeLa cells overexpressing ZDHHC6 mutants. HeLa cells were transfected with control plasmid (Ctrl) or plasmids encoding WT or mutants myc-ZDHHC6 for 24 hr. Cells were then metabolically labelled 2 hr at 37°C with $^3$H-palmitic acid. Proteins were extracted and immunoprecipitated with anti-calnexin or anti-Trf-R antibodies, subjected to SDS-PAGE and analysed by autoradiography ($^3$H-palm), quantified using the Typhoon Imager. A representative experiment is shown for the Trf-R (G). $^3$H-palmitic acid incorporation into calnexin or Trf-R was set to 100% for the control plasmid and all ZDHHC6 mutants were expressed relative to this (n = 4, errors represent standard deviations).

DOI: https://doi.org/10.7554/eLife.27826.010

The following figure supplement is available for figure 4:

**Figure supplement 1.** Immunofluorescence staining of HeLa cells transfected with myc-DHHC6 WT or cysteine mutants.
DOI: https://doi.org/10.7554/eLife.27826.011

Certain DHHC enzymes were reported to dimerize (*Lai and Linder, 2013*). We therefore tested whether ZDHHC6 also forms higher order structures, and if so whether assembly is influenced by palmitoylation. Immunoprecipitation experiments using ZDHHC6 constructs with two different tags showed that WT ZDHHC6 can associate with itself as well as with the ΔDHHC variant but not with the AAA mutant (*Figure 4C*). Blue native gel analysis confirmed that ZDHHC6 can associate into higher order structures, which could be dimers and above, or associate with detergent resistant membrane domains (*Figure 4D*). In contrast DHHC 7 and 20 appeared largely monomeric under the same solubilization conditions (*Figure 4D*). The AAA mutant also migrated as higher order complexes but these were clearly different from those formed by the WT protein (*Figure 4D*). Thus palmitoylation affects localization of ZDHHC6 in the ER as well as its assembly into complexes or association with membrane domains.

## Palmitoylation controls ZDHHC6 activity

Finally we tested whether palmitoylation of ZDHHC6 affects its ability to modify reported substrates in a cellular context. Calnexin palmitoylation was significantly reduced in *ZDHHC16* KO cells, as monitored by Acyl-RAC (*Figure 4E*) and enhanced upon ZDHHC16 over-expression (*Figure 4F*) indicating that palmitoylation indeed modulates ZDHHC6 activity. We next monitored $^3$H-palmitate incorporation into endogenous transferrin receptor and calnexin upon overexpression of WT and ZDHHC6 mutants in HeLa cells. Over-expression of WT ZDHHC6 led to 2.6 to 3-fold increase in calnexin and transferrin receptor palmitoylation, respectively (*Figure 4GH*). The background level of palmitoylation is due to the presence of endogenous ZDHHC6 (*Figure 4GH*). Overexpression of ZDHHC6 AAA had no effect on either target, confirming the importance of palmitoylation for ZDHHC6 activity. Overexpression ACC, AAC and ACA ZDHHC6 mutants barely affected Trf-R and calnexin palmitoylation, as opposed to the CAA, CCA and CAC mutants (*Figure 4GH*). The fact that palmitoylation of both ZDHHC6 targets is similarly affected by cysteine mutations rules out the possibility that differentially palmitoylated ZDHHC6 molecules could have different target specificities. Together these experiments indicate that palmitoylation of Cys-328 confers the highest activity to ZDHHC6.

## Model of the ZDHHC6 palmitoylation system

The results presented in the previous paragraphs show that ZDHHC6 can be palmitoylated on three sites, that the protein undergoes cycles of palmitoylation-depalmitoylation and that palmitoylation strongly influences assembly, localization, stability and function of the enzyme. The presence of 3

palmitoylation sites leads to the potential existence of 8 species: from fully unoccupied Cys-sites, noted $C^{000}$, to full occupancy, noted $C^{111}$, in the order of Cys-328, Cys-329 and Cys-343 in the exponent (**Figure 5A**). To understand the dynamics of the inter-conversion between the eight species, as well as derive hypothesis about the role of the three sites, we developed a mathematical model (**Supplementary file 1A**). Modelling was performed as an open system, including protein synthesis, and degradation of all species (**Figure 5A** and **Supplementary file 1B**). Synthesis of ZDHHC6 first leads to an unfolded species (U). Folded ZDHHC6 protein is initially in the $C^{000}$ form, which can undergo palmitoylation on any of the three sites leading to $C^{100}$, $C^{010}$, $C^{001}$. These single palmitoylated species can undergo a second and a third palmitoylation event first leading to $C^{110}$, $C^{101}$, $C^{011}$ and finally to $C^{111}$. Each palmitoylation reaction is mediated by ZDHHC16 and each depalmitoylation catalysed by APT2. A competition term between the sites was implemented in the enzymatic kinetics, as previously for the modelling of palmitoylation of the ER chaperone calnexin (**Dallavilla et al., 2016**). The model includes first-order degradation rates for each species, with different rate constants. The rate expressions, the parameters and the assumptions used in the development of the model are described in detail in the Materials and methods and the Supplementary Information (**Supplementary file 1A**).

With the aim of first calibrating and subsequently validating the model, we generated the following kinetic datasets. We performed: (1) metabolic $^{35}$SCys/Met pulse-chase experiments, with pulses of either 20 min or 2 hr, to monitor the turnover of newly synthesized proteins, for WT and the different single, double and triple cysteine mutants; (2) $^3$H palmitate incorporation into WT and cysteine mutants; (3) palmitate loss, monitored by $^3$H palmitate pulse-chase experiments, for WT and mutants. Some of these experiments were in addition performed upon over expression of ZDHHC16 or siRNA of *LYPLA2*.

We used the $^{35}$S-decay data for WT, ACC, CAA and AAA, the WT $^3$H-palmitate incorporation data and the $^3$H-palmitate release data for WT, ACC and CAA, to calibrate the model. The pulse labelling times were included in the model. During parameter estimation, these calibration datasets were fitted simultaneously, as multi-objective optimization problem with as many fitness functions as experiments. As often for optimization problems, the output of the algorithm is a local Pareto set of solutions. These are equally optimal with respect to the fitness functions in the sense that for each set of parameters, none of the objective functions can be improved in value without deteriorating the quality of the fitness for some of the other objective values. We therefore employed a stochastic optimization method to generate a population of models consistent with the calibration experiments. From a population of 10'000 models, we selected 152 based on the scores of the objective functions that fitted the experimental data most accurately (**Figure 5—figure supplement 1**). The pool of selected models was subsequently used for the simulations and analyses. Importantly, all predictions were obtained by simulating each model independently. Outputs of all models were averaged and standard deviations with respect to the mean were calculated (**Figure 5B**).

The remaining experiments, not used to calibrate the model, were used to validate it. Importantly, as can be seem in **Figure 5B** and **Figure 5—figure supplement 2**, the model reliably predicted the results of these experiments, indicating that it accurately captures the ZDHHC6 palmitoylation system.

## Dynamics and consequences of site-specific ZDHHC6 palmitoylation

The model was first used to estimate the half-lives of the different species. $C^{000}$ is predicted to have a half-life of approx. 40 hr (**Figure 5C** and **Supplementary file 2A**), consistent with the decay measured experimentally for the AAA triple mutant (**Figure 5—figure supplement 2**). The presence of palmitate on Cys-328 is predicted to strongly accelerate protein turnover, irrespective of the occupancy of the other sites, with $t_{1/2}$=5 hr for $C^{100}$ and $t_{1/2}$=0.3 h $C^{111}$ (**Figure 5C** and **Supplementary file 2A**). In contrast, palmitate on Cys-329 is predicted to have a stabilizing effect with $t_{1/2}$>100 hr for $C^{010}$. Finally, palmitate on Cys-343 would have a moderately destabilizing effect with $t_{1/2}$=18 hr for $C^{001}$ (**Figure 5C**). Thus, the model indicates that palmitoylation strongly affects ZDHHC6 turnover in a site dependent manner. It is interesting to note that during the course of this study during which we performed iterative cycles of experiments and modelling, which led to the current model, the mathematical approach provided the initial hint that palmitoylation of Cys-328 had a stabilizing effect. This was not visible from our initial 20 min metabolic pulse experiments of

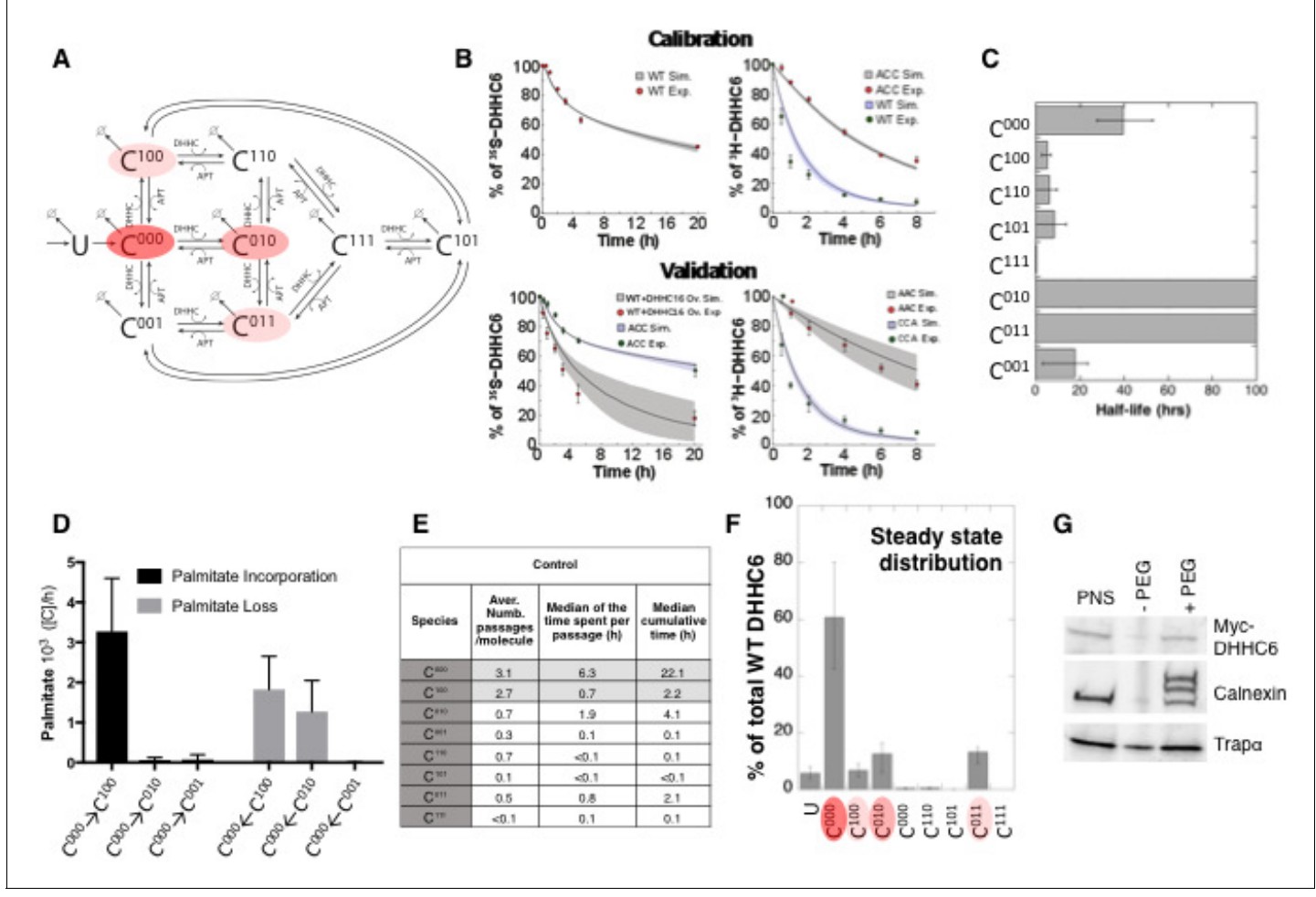

**Figure 5.** Modelling and analysis of ZDHHC6 palmitoylation. (**A**) Network topology of the ZDHHC6 palmitoylation model. First we have a phase of synthesis of the unfolded peptide (U). The protein goes through a process of folding and membrane embedding, ending in the fully folded form ($C^{000}$). The three sites can then be palmitoylated by ZDHHC16, the first palmitoylation can occur on each site, $C^{100}$ $C^{010}$ and $C^{001}$ denote palmitoylation on the first, second or third site respectively. ZDHHC6 can then undergo another palmitoylation step acquiring two palmitates; $C^{110}$, $C^{101}$ and $C^{011}$ denote the double palmitoylated enzyme. From each of the three double palmitoylated states, ZDHHC6 can be modified one last time becoming fully palmitoylated ($C^{111}$). We consider that ZDHHC6 degradation can happen in any state. Highlighted in red are the most abundant species in steady state. (**B**) Part of the calibration and validation sets used for parameter estimation. In the top graphs we show three curves that were used during parameter estimation with genetic algorithm to evaluate the goodness of fit of the parameters generated. Other curves used for the validation are shown in *Figure 5—figure supplement 1*. The four curves in the bottom graphs are part of the data that were used for validating the predictions on experiments that were not used for parameter calibration. Circles represent experimental data while the solid lines are the output of the model after optimization. Since we have 152 different sets of optimal parameters, the shadows behind the lines represent the first and third quartile of the 152 model outputs. All other curves used for parameter estimation can be found in *Figure 5—figure supplement 2*. (**C**) Half-life of the different ZDHHC6 palmitoylation states estimated from the decay rate constants of the model after optimization. The half-life was calculated as: $\ln(2)/kd_i$. Where $kd_i$ is the decay rate constant of the i-th palmitoylation state. These are not experimentally determined apparent turn over rates. (**D**) Palmitoylation and depalmitoylation fluxes for the six steps of single palmitoylation/depalmitoylation in steady state, which represent the fluxes of palmitate incorporation and loss on the first, second and third site during the first palmitoylation event. Fluxes in steady state for all reactions are shown in *Figure 5—figure supplement 3*. (**E**) Single molecule tracking with stochastic simulations. The table shows the average number of passage per molecule for each state of the model, along with the median and the cumulative median of the time spent in each state. These data were obtained from the analysis of 10'000 stochastic simulations. (**F**) Prediction of the steady state distribution of ZDHHC6 WT palmitoylation species. (**G**) Stoichiometry of ZDHHC6 palmitoylation in HeLa cells. HeLa cells were transfected with plasmids encoding WT myc-ZDHHC6 constructs for 24 hr. Protein lysates were processed for the APEGS assay. PEG-5k was used to label transfected myc-ZDHHC6 and endogenous protein (PEG+), PEG- lanes indicate the negative controls. The samples were analysed by western blotting with anti-myc, anti-calnexin, anti-TRAP alpha antibodies.

DOI: https://doi.org/10.7554/eLife.27826.012

The following figure supplements are available for figure 5:

**Figure supplement 1.** Calibration of the ZDHHC6 palmitoylation model.

*Figure 5 continued on next page*

*Figure 5 continued*

DOI: https://doi.org/10.7554/eLife.27826.013
**Figure supplement 2.** Validation.
DOI: https://doi.org/10.7554/eLife.27826.014
**Figure supplement 3.** Model fluxes in steady state.
DOI: https://doi.org/10.7554/eLife.27826.015

WT and cysteine mutants. The model predicted that increasing the pulse labelling time to 2 hr should make the difference between WT ZDHHC6 and mutants turnover apparent, which turned out to be correct. We subsequently tested the effect of ZDHHC16 overexpression and *LYPLA2* silencing, providing further confirmation. Thus the evolving model led to experiments that were key for proper parameter estimation and for deeper understanding of the system.

We next analysed the dynamics of the system. We first determined the palmitoylation and depalmitoylation fluxes. The major fluxes through the system are from $C^{000}$ to $C^{100}$, backwards, corresponding to palmitoylation and depalmitoylation of Cys-328 (*Figure 5D*). Depalmitoylation of Cys-329 is also predicted as quite prominent (*Figure 5D*).

In order to quantify how much time each protein molecule spends in the different states and the transition dynamics between the different states, we derived a stochastic formulation of the model, as we have previously done for calnexin palmitoylation (*Dallavilla et al., 2016*) (Suppl. Information, *Supplementary file 3*). While the deterministic simulation could provide an estimate of these properties, the stochastic simulations in addition provides a quantification of the distribution of the times of interest and allow asking questions about 'individual' molecules. We performed 10'000 simulations to track single proteins in the system. ZDHHC6 molecules were predicted to spend most of their time, by far, in the $C^{000}$ state (*Figure 5E*). Consistent with the flux analysis, each ZDHHC6 molecule was on average 2.7 times in the $C^{100}$ state (*Figure 5E*), and remained in that state for about 0.7 hr (median value). Seven out of 10 molecules also explored the $C^{010}$ state for almost 2 hr and more briefly the $C^{011}$ state.

Estimation of the steady state distribution of species indicated that in experimental setting –HeLa cells in culture– the ZDHHC6 cellular population is about 70% non-palmitoylated and 20% palmitoylated, the most abundant species being $C^{010}$ and $C^{011}$ (*Figure 5F*). We tested this prediction experimentally by performing a PEGylation assay. This is a mass-tag labelling method that consists in replacing the palmitate moiety with PEG following disruption of the thioester bond with hydroxylamine, resulting in a mass change detectable by electrophoresis and Western blotting (*Howie et al., 2014*; *Percher et al., 2016*; *Yokoi et al., 2016*). As a control we analysed calnexin, which has two palmitoylation sites and migrated as expected as three bands, corresponding to the non-, single and dual palmitoylated forms (*Figure 5G*). For ZDHHC6, only the non-palmitoylated form was detected, consistent with the prediction (*Figure 5FG*). Indeed given the dynamic range and sensitivity of Western blots, a band with a ≈7-fold lower intensity than the $C^{000}$ ZDHHC6 band would not be detectable.

The model altogether predicts that in tissue culture HeLa cells, each ZDHHC6 molecule undergoes multiple rounds of palmitoylation-depalmitoylation during its life cycle mostly on Cys-328. Only a subset of molecules acquires palmitate on the two other sites. At any given time, approximately 60% of the molecules are not acylated.

## Multi-site palmitoylation protects ZDHHC6 from degradation

Analysis of the activity of the ZDHHC6 mutants (*Figure 4H*) showed that the most active variants are those with a cysteine at position 328, suggesting that palmitoylation at this position influences palmitoyltransferase activity. However palmitoylation at this site also targets ZDHHC6 to ERAD (*Figure 3D*). These effects raise the question of how cells can increase cellular ZDHHC6 activity.

We first predicted the consequences of ZDHHC6 hyperpalmitoylation. Calculations were performed under conditions of ZDHHC16 overexpression and of *Lypla2* silencing, simulated by an absence of APT2. ZDHHC16 overexpression led to a major shift in species distribution, $C^{011}$ then representing 60% of the population (*Figure 6A*). Interestingly, the overall ZDHHC6 content dropped by only 15% (*Supplementary file 2B*). *Lypla2* silencing in contrast led to a 72% drop in ZDHHC6 content (*Supplementary file 2B*). Since removing all depalmitoylating activity is likely to freeze the

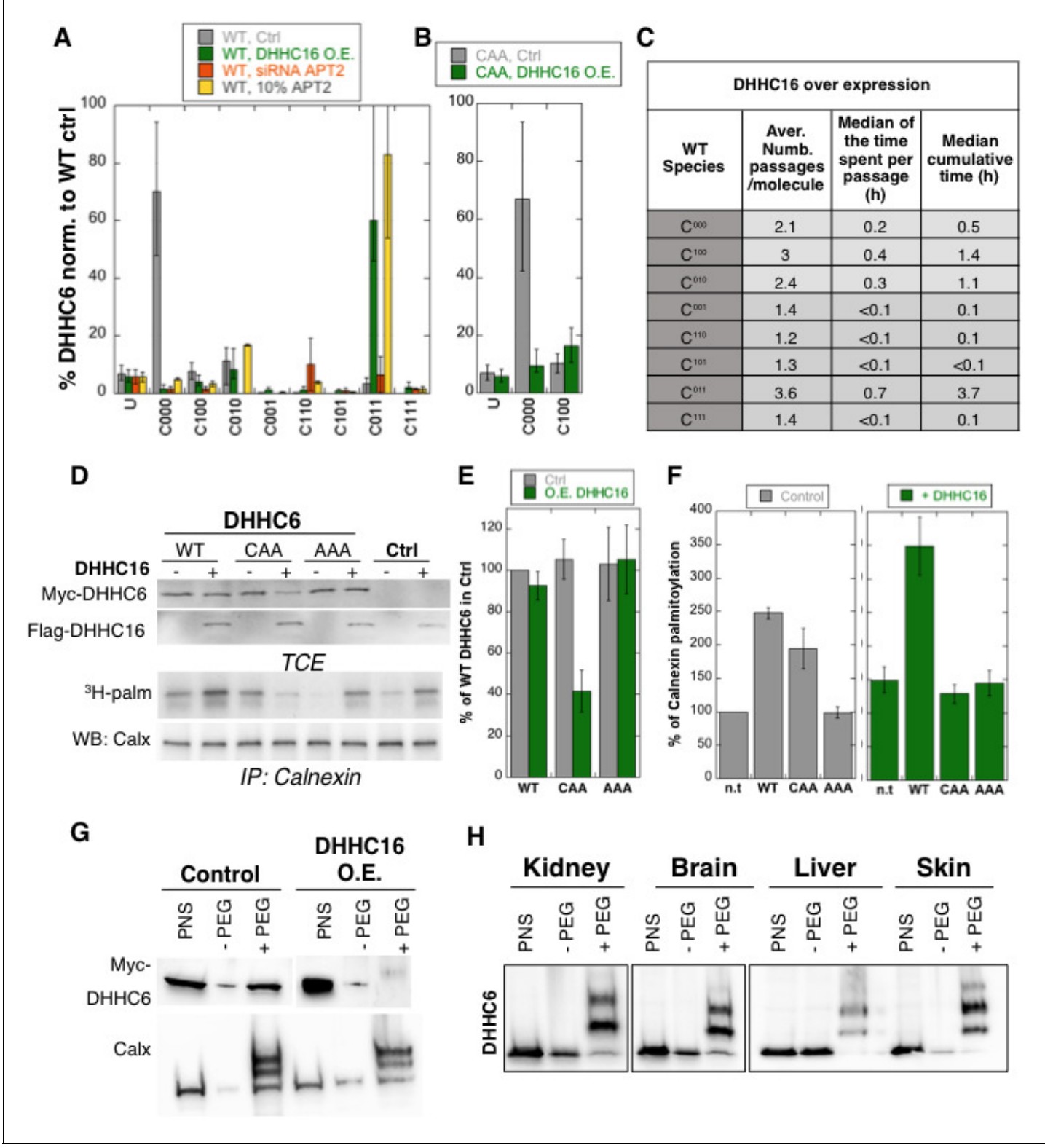

**Figure 6.** Importance of multiple palmitoylation sites. (A) Single molecule tracking with stochastic simulations upon ZDHHC16 overexpression. The table shows the average number of passage per molecule in each state of the model, along with the median and the cumulative median of the time spent in each state. As for control experiments this numbers were obtained by averaging the results of 10'000 independent stochastic simulations. (B) Steady state ZDHHC6 WT species distribution under different conditions (control, ZDHHC16 Overexpression, *LYPLA2* (gene encoding APT2) silencing and APT2 at 10% level of expression with respect to WT condition). All the data are scaled with respect to the total abundance of ZDHHC6 WT under

*Figure 6 continued on next page*

*Figure 6 continued*

control condition. (C) Steady state distribution of the ZDHHC6 CAA mutant under control condition or ZDHHC16 overexpression. All the data are scaled with respect to the total abundance of ZDHHC6 WT in normal condition. (D) Calnexin palmitoylation upon ZDHHC6 and ZDHHC16 overexpressed. HeLa cells were transfected with control plasmid (Ctrl) or plasmids encoding WT or mutants myc-ZDHHC6 in the presence or not of FLAG-ZDHHC16 for 24 hr. Cells were then metabolically labelled 2 hr at 37°C with $^3$H-palmitic acid. Proteins were extracted, TCE isolated, and immunoprecipitated with anti-calnexin antibodies, subjected to SDS-PAGE and analysed by autoradiography ($^3$H-palm), quantified using the Typhoon Imager or by immunoblotting with anti-calnexin, anti-myc or anti-flag antibodies. (E) Quantification of ZDHHC6 levels in total cell extracts with or without ZDHHC16 overexpression. The calculated value for WT ZDHHC6 under control condition was set to 100%. All ZDHHC6 mutants with or without ZDHHC16 were expressed relative to this (n = 3, errors represent standard deviations). (F) Quantification of $^3$H-palmitic acid incorporation into endogenous calnexin with ZDHHC6 in the absence or presence of ZDHHC16. The calculated value of $^3$H-palmitic acid incorporation into calnexin was set to 100% for control plasmid (Ctrl) and all ZDHHC6 mutants were expressed relative to this, n = 3. (G) Stoichiometry of ZDHHC6 palmitoylation in HeLa cells with ZDHHC16 overexpressed. HeLa cells were transfected with plasmids encoding WT myc-ZDHHC6 with Flag-ZDHHC16 constructs for 24 hr. Protein lysates were processed for the APEGS assay. PEG-5k were used for labelling of transfected myc-ZDHHC6 (PEG+), PEG- lanes indicate the negative controls. The samples were analysed by western blotting with anti-myc. (H) Adult mouse tissues were processed for the APEGS assay. PEG-5k were used for labelling of endogenous ZDHHC6 (PEG+), PEG- lanes indicate the negative controls. The samples were analysed by western blotting with rabbit anti-ZDHHC6.

DOI: https://doi.org/10.7554/eLife.27826.016

system (*Supplementary file 4*), we also tested a decrease of APT2% to 10% of normal levels. The ZDHHC6 content rose by 20% and, as upon ZDHHC16 overexpression, $C^{011}$ became the most abundant form (*Figure 6A*). We made similar calculations for the CAA mutant. Under control conditions, the total CAA content is predicted to be 84% of WT, and mostly in the non-palmitoylated state (*Figure 6B*). The CAA cellular content is however predicted to drop to 32% of WT control levels upon ZDHHC16 overexpression (*Figure 6B*). All together, these predictions suggest that the presence of 3 palmitoylation sites, as opposed to just Cys-328, renders the ZDHHC6 protein content robust to changes in ZDHHC16 activity.

Stochastic simulations predicted that overexpression of ZDHHC16 would drastically increase the dynamics of the network (*Figure 6C* vs. *Figure 5E*). Indeed, all ZDHHC6 molecules explored all the palmitoylation states, with extremely short residence times in each (*Figure 6C*). Interestingly, flux analysis indicated that the abundance of $C^{011}$ was primarily due to two 4-step paths: both started with $C^{000}$ to $C^{100}$, followed by palmitoylation on either of the two remaining sites, subsequent depalmitoylation of Cys-328, and finally palmitoylation of the remaining site (*Figure 5—figure supplement 3*). When analysing 10'000 simulations, 22'000 events of palmitoylation-depalmitoylation emanated from $C^{011}$ whereas only 4'000 events occurred between $C^{000}$ and $C^{100}$. Thus when activity of ZDHHC16 is high, $C^{011}$ appears to be the hub of the system. $C^{011}$ has the slowest turover rate of all ZDHHC6 species, explaining why the protein levels are stable.

We next sought to validate these predictions experimentally. Western blot analysis of protein abundance showed that WT ZDHHC6 protein levels drastically dropped upon siRNA of *Lypla2* (*Figure 4B*), but not upon ZDHHC16 overexpression (*Figure 6DE*). ZDHHC16 overexpression however led to a 60% drop in CAA expression (*Figure 6DE*). As expected, expression of AAA was not affected (*Figure 6DE*).

## Multi-site palmitoylation regulates ZDHHC6 activity

We next analysed the importance of multiple palmitoylation sites on the ability of ZDHHC6 to modify it substrates. WT or CAA ZDHHC6 were ectopically expressed, or not, in HeLa cells, which also express endogenous ZDHHC6. In addition, ZDHHC16 was ectopically or not, on top of the endogenous ZDHHC16. Under control ZDHHC16 levels, both ectopic expression of WT and CAA ZDHHC6 led to an increase in calnexin palmitoylation (*Figure 6D–F* as already shown in *Figure 4H*. The ability of WT ZDHHC6 to modify calnexin was enhanced by ZDHHC16 overexpression (*Figure 6D–F*). In contrast, ZDHHC16 overexpression reduced calnexin palmitoylation to background levels in CAA ZDHHC6 expressing cells (*Figure 6D–F*). Note that the 'background' calnexin palmitoylation level in the right panel of *Figure 6F* was increased compared to the left panel due to the activation of endogenous ZDHHC6 by ZDHHC16 over expression. Using PEGylation, we confirmed that ZDHHC16 overexpression leads to an increase of the palmitoylated ZDHHC6 species (*Figure 6H*).

The above experimental analyses show that in HeLa cells under control tissue culture conditions, ZDHHC6 is mainly in the non-palmitoylated form. When ZDHHC6 PEGylation experiments were performed on different mouse tissues, we however found that ZDHHC6 is far more palmitoylated, suggesting that palmitoylation-mediated ZDHHC6 activity is higher in vivo, a situation that we could mimic in cells by ZDHHC16 overexpression.

## Discussion

S-Palmitoylation is a reversible lipid modification that can control protein function in time and in space. The enzymes involved must therefore also be regulated. The DHHC family of palmitoyltransferases was identified almost 15 years ago, yet little is known about the mechanisms that control their activity. DHHC9 was found to require a co-factor, GCP16, for activity (*Swarthout et al., 2005*), as also found in yeast (*Lobo et al., 2002*). Similarly, ZDHHC6 was proposed to require Selenoprotein K to function (*Fredericks and Hoffmann, 2015*). Here we show that ZDHHC6 activity is controlled by palmitoylation in its C-terminal SH3_2-domain. In the non-palmitoylated form, ZDHHC6 has no detectable activity. Only upon modification by an upstream palmitoyltransferase, which we identified as ZDHHC16, does it acquire significant transferase activity. ZDHHC16, also known as Ablphilin 2 (Aph2), is, as ZDHHC6, expressed in many human tissues in particular in the heart, pancreas, liver, skeletal muscle (*Zhang et al., 2006*). Its name Ablphilin stems from its ability to interact with the non-receptor tyrosine kinase c-Abl at the ER surface (*Li et al., 2002*). ZDHHC16/Aph2 is essential for embryonic and postnatal survival, eye and heart development in mice (*Zhou et al., 2015*), for the proliferation of neural stem cells, where it is involved in the activation of the FGF/erk pathways (*Shi et al., 2016*) and was reported to play a role in DNA damage response (*Cao et al., 2016*). Our findings raise the possibility that some of these effects may involve ZDHHC6.

We found that ZDHHC6 palmitoylation is highly dynamic, involving Acyl Protein Thioesterase 2. APT2, which was recently found to be involved in cell polarity-mediated tumor suppression (*Hernandez et al., 2017*) but is otherwise poorly characterized, also undergoes palmitoylation by a yet to be determined palmitoyltransferase (*Kong et al., 2013*; *Vartak et al., 2014*).

ZDHHC6 palmitoylation can occur on three sites in its SH3_2 domain. We combined mathematical modelling, with mutagenesis of palmitoylation sites, expression levels of the involved enzymes (ZDHHC16 and APT2) and experimental determination of palmitoylation, depalmitoylation, metabolic pulse-chase experiments and activity determination, to understand the importance of the different ZDHHC6 palmitoylation species and the dynamics of their interconversion. In addition to the predictive power of mathematical modelling of biological systems, this study provides single molecule understanding, as opposed to population analysis.

We found that in tissue culture cells, under standard conditions, ZDHHC6 molecules spend more than 70% of their lifetime in the non-palmitoylated inactive $C^{000}$ state (*Figure 5E*). Each ZDHHC6 molecule does however undergo several rounds of palmitoylation-depalmitoylation on Cys-328 (*Figure 5E*), leading to $C^{100}$, the most active state of the enzyme. According to large-scale quantitative proteomics analyses, DHHC enzymes are typically present in less than 1000 copies per cell (*Beck et al., 2011*; *Foster et al., 2003*; *Nagaraj et al., 2011*). Together this information indicates that, at any given time, tissue culture cells contain a very low number of active ZDHHC6 enzymes. In these cells, target proteins do however undergo palmitoylation, and some of these targets are highly abundant proteins such as calnexin (half a million to a million copies per cell [*Beck et al., 2011*; *Foster et al., 2003*; *Nagaraj et al., 2011*]). Thus when active, ZDHHC6 appears be a very potent enzyme and our observations indicate that cells tend to avoid having too much of it. Indeed the $C^{100}$ species of ZDHHC6 is either rapidly depalmitoylated or targeted to degradation via the ERAD pathway.

PEGylation analysis of mouse tissues indicate that in vivo ZDHHC6 palmitoylation is far more pronounced than in tissue cultured cells (*Figure 6H*). This situation was mimicked experimentally by overexpression of ZDHHC16 and revealed the importance of having 3 palmitoylation sites, rather than just Cys-328. We indeed found that in the presence of 3 sites, the cellular ZDHHC6 activity could be enhanced by ZDHHC16 overexpression. Increase ZDHHC16 activity led to a shift of species distribution: $C^{011}$, the most stable form, became the most abundant species. Altogether this study indicates that ZDHHC6 can adopt in 3 types of states: 1) stable and inactive ($C^{000}$); 2) highly active, short-lived and rapidly turned over ($C^{100}$); 3) moderately active, long-lived and very slowly turned

over ($C^{010}$ and $C^{011}$). Such a regulatory system allows the cell to tightly control the activity of ZDHHC6. Why excessive ZDHHC6 activity is detrimental to a cell or an organism will require further studies.

By combining experimentation and the predictive power of data-driven mathematical modelling, we were able to obtain unprecedented insight into the dynamics of palmitoylation and the role and properties of single acylated species. We found that depending on the site of modification both the activity and turnover of ZDHHC6 were strongly affected, indicating that palmitoylation has a more subtle and precise effect than merely increasing hydrophobicity, consistent with recent findings on single acylated species of Ras (*Pedro et al., 2017*). We moreover revealed that palmitoylation can occur in cascades, ZDHHC16 controlling the activity of ZDHHC6, which itself tunes the activity of key proteins of the ER and other proteins of the endomembrane system. Finally, this study highlights the importance of acyl protein thioesterases in regulating the interconversion between palmitoylated species.

# Materials and methods

## Cell lines

HeLa cells (ATCC) were grown in complete modified Eagle's medium (MEM, Sigma) at 37°C supplemented with 10% foetal bovine serum (FBS), 2 mM L-Glutamine, penicillin and streptomycin. HeLa cells are not on the list of commonly misidentified cell lines maintained by the International Cell Line Authentication Committee. Our cells were authenticated by Microsynth (Switzerland), which revealed 100% DNA identity with ATCCCCL-2. They were mycoplasma negative as tested on a trimestral basis using the MycoProbe Mycoplasma Detection Kit CUL001B. For the *ZDHHC6* knockdown cell lines, HeLa cells were transfected with shRNA against *ZDHHC6* gene (target sequence in 3'UTR: 5'-CCTAGTGCCATGATTTAAA-3') or with shRNA control against firefly luciferase gene (target sequence: 5'-CGTACGCGGAATACTTCGA-3'). The transfected cells were selected by treatment with 3 µg/ml puromycin. HAP1 Wild type WT and knockout cell lines were purchased from Horizon Genomics (Vienna, Austria). The *ZDHHC6* clone (13474–01) contains a 5 bp deletion in exon 2 (NM_022494) and the *ZDHHC16* clone (36523–06) contains a 2 bp insertion in exon 2 (NM_198043). HAP1 cells were grown in complete Dulbeccos MEM (DMEM, Sigma) at 37°C supplemented with 10% foetal bovine serum (FBS), 2 mM L-Glutamine, penicillin and streptomycin.

## Antibodies and reagents

The following primary antibodies are used: Mouse anti-Actin (Millipore, MAB 1510), Mouse anti-Myc 9E10 (Covance, MMs-150R), Mouse anti-Ubiquitin (Santa Cruz, sc-8017), Mouse anti-GFP (Roche, 11814460001), Rabbit anti-ZDHHC6 (Sigma, SAB1304457), Mouse anti-Transferrin Receptor (Thermo Scientific, 136800), Mouse anti-Flag M2 (Sigma, F3165), Rabbit anti-ANTXR1 (Sigma, SAB2501028), Rabbit anti-TRAPα (Abcam, ab133238), Rabbit anti-Flotillin1 were produced in our laboratory, Mouse anti-CLIMP63 (Enzo, ALX-804–604), Mouse anti-Calnexin (MAB3126), Rabbit anti-GP78 AMFR (Abnova, PAB1684), Rabbit anti-IP3R (Cell signaling, 85685). The following beads were used for immunoprecipitation: Protein G Sepharose 4 Fast flow (GE Healthcare, 17-0618-01), anti-Myc affinity gel (Thermo Scientific, 20169), anti-Flag affinity gel EZview M2 (Sigma, F2426). Drugs were used as follows: Bafilomycin A1 at 100 nM (Sigma, B1793), MG132 at 10 µM (Sigma, C2211), Hydroxylamine at 0.5 M (Sigma, 55459), mPEG-5k at 20 mM (Sigma, 63187), N-ethylmaleimide NEM at 20 mM (Thermo Scientific, 23030), Tris-2- carboxyethyl-phosphine hydrochloride TCEP at 10 mM (Thermo Scientific, 23225), Methyl methanethiosulfonate MMTS at 1.5% (Sigma, 208795), Puromycin at 3 µg/ml (Sigma, P9620).

## Transfection and siRNA experiments

Human Myc-ZDHHC6, Myc-ZDHHC6- C328A (ACC), Myc-ZDHHC6-C329A (CAC), Myc-ZDHHC6-C343A (CCA), Myc-ZDHHC6-C328A-C329A (AAC), Myc-ZDHHC6-C326A-C343A (ACA), Myc-ZDHHC6-C329A-C343A (CAA), Myc-ZDHHC6-C328A-C329A-C343A, Myc-ZDHHC6-R361Q, Myc-ZDHHC6-R361A, Myc-ZDHHC6-Del-K410-K411-N412-R413, Myc-ZDHHC6-Del-D126-H127-H128-C129 were cloned in pcDNA3. Human GFP-ZDHHC6 and all cysteine mutants mentioned above were also cloned in peGFP. Human FLAG-ZDHHC16 was cloned in pCE-puro-3xFLAG, human Myc-

ZDHHC16 was cloned in pCE puro-his-myc. All other human Myc-DHHCs were cloned in pcDNA3.1 (provided by the Fukata lab). mCitrine fusions of APTs were inserted into pcDNA3.1-N1 (provided by Bastiaens lab, [Vartak et al., 2014]). mCitrine APT1-S119A, mCitrine APT2-S121A, mCitrine-APT1-C2S, mCitrine-APT2-C2S were cloned in pcDNA3.1-N1. For control transfection, we used an empty pcDNA3 plasmid. Plasmids were transfected into HeLa cells for 24 hr (2 µg cDNA/9.6 cm$^2$) plate using Fugene (Promega).

For gene silencing, HeLa cells were transfected for 72 hr with 100 pmol/9.2cm2 dish of siRNA using interferin (Polyplus) transfection reagent. As control siRNA we used the following target sequence of the viral glycoprotein VSV-G: ATTGAACAAACGAAACAAGGA. siRNA against human genes were purchased from Siegen (ZDHHC6 target sequences: 1-GAGGTTTACGATACTGGTTAT, 2-TAGAAGGTGTTTCAAGAATAA, ZDHHC16 target sequences: 1-CTCGGGTGCTCTTACCTTCTA, 2-TAGCATCGAAAGGCACATCAA; APT1 target sequence: AACAAACTTATGGGTAATAAA; LYPLA2 target sequence: AAGCTGCTGCCTCCTGTCTAA, all other DHHC target sequences were previously tested [Lakkaraju et al., 2012]).

## Real-time PCR

For HeLa cells, RNA was extracted from a six-well dish using the RNeasy kit (Qiagen, Hilden, Germany). 1 mg of the total RNA extracted was used for the reverse transcription using random hexamers and superscript II (Thermo Scientific). A 1:40 dilution of the cDNA was used to perform the real-time PCR using SyBr green reagent (Roche). mRNA levels were normalized using three housekeeping genes: TATA-binding protein, β-microglobulin and β-glucoronidase. Total RNA of different mouse tissues were extracted using the RNeasy kit (Qiagen) after solubilization with TissueLyser II (Qiagen).

## Radiolabelling experiments

For the $^{35}$S-metabolic labelling, the cells were starved in DMEM HG devoid of Cys/Met for 30 min at 37°C, pulsed with the same medium supplemented with 140 µCi of $^{35}$S Cys/Met (American Radiolabeled Chemicals, Inc.) for the indicated time, washed and incubated in DMEM complete medium for the indicated time of chase (Abrami et al., 2008) before immunoprecipitation. To detect palmitoylation, HeLa cells were transfected or not with different constructs, incubated for 2 hr in IM (Glasgow minimal essential medium buffered with 10 mM Hepes, pH 7.4) with 200 µCi/ml $^3$H palmitic acid (9,10-$^3$H(N)) (American Radiolabeled Chemicals, Inc.). The cells were washed, incubated in DMEM complete medium for the indicated time of chase, or directly lysed for immunoprecipitation with the indicated antibodies.

For all radiolabelling experiments, after immunoprecipitation, washes beads were incubated for 5 min at 90°C in reducing sample buffer prior to 4–12% gradient SDS-PAGE. After SDS-PAGE, the gel are incubated in a fixative solution (25% isopropanol, 65% H2O, 10% acetic acid), followed by a 30 min incubation with signal enhancer Amplify NAMP100 (GE Healthcare). The radiolabeled products were revealed using Typhoon phosphoimager and quantified using the Typhoon Imager (Image-QuanTool, GE Healthcare). Quantification of radioactive experiments was quantified using specific screens by autoradiography, both for $^{35}$S and $^3$H. The images shown for $^3$H-palmitate labeling were however obtained using fluorography (indirect detection of radioactive emission by stimulated light emission from a fluorophore (Amplify) on film).

## Immunoprecipitation

For immunoprecipitation, cells were washed three times PBS, lysed 30 min at 4°C in the following Buffer (0.5% Nonidet P-40, 500 mM Tris pH 7.4, 20 mM EDTA, 10 mM NaF, 2 mM benzamidin and protease inhibitor cocktail (Roche)), and centrifuged 3 min at 5000 rpm. Supernatants were subjected to preclearing with G sepharose beads prior immunoprecipitation reaction. Supernatants were incubated overnight with the appropriate antibodies and G Sepharose beads.

## Post-nuclear supernatants and ACYL-RAC

HeLa cells were harvested, washed with PBS, and homogenized by passage through a 22G injection needle in HB (HB: 2.9 mM imidazole and 250 mM sucrose, pH 7.4) containing a mini tablet protease

inhibitor mixture (Roche). After centrifugation, the supernatant was collected as PNS (Post Nuclear Supernatant).

Protein S-palmitoylation was assessed by the Acyl-RAC assay as described (*Werno and Chamberlain, 2015*), with some modifications. A fraction of the PNS was saved as the input. HeLa PNS were lysed in buffer (0.5% Triton-X100, 25 mM HEPES, 25 mM NaCl, 1 mM EDTA, pH 7.4 and protease inhibitor cocktail). In order to block free SH groups with S-methyl methanethiosulfonate (MMTS), 200 µl of blocking buffer (100 mM HEPES, 1 mM EDTA, 87.5 mM SDS and 1.5% (v/v) MMTS) was added to cell lysate and incubated for 4 h at 40 °C. Subsequently, 3 volumes of ice-cold 100% acetone was added to the blocking protein mixture and incubated for 20 minutes at −20 °C and then centrifuged at 5,000 × g for 10 minutes at 4 °C to pellet precipitated proteins. The pellet was washed five times in 1 ml of 70% (v/v) acetone and resuspended in buffer (100 mM HEPES, 1 mM EDTA, 35 mM SDS). For treatment with hydroxylamine (HA) and capture by Thiopropyl Sepharose beads, 2 M hydroxylamine was added together with the beads (previously activated for 15 min with water) to a final concentration of 0.5 M hydroxylamine and 10% (w/v) beads. As a negative control, 2 M Tris was used instead of hydroxylamine. These samples were then incubated overnight at room temperature on a rotate wheel. The beads were washed, the proteins were eluted from the beads by incubations in 40 µl SDS sample buffer with beta-mercapto-ethanol for 5 minutes at 95 °C. Finally, samples were submitted to SDS-PAGE and analysed by immunoblotting.

### APEGS assay

The level of protein S-palmitoylation was assessed as described (*Yokoi et al., 2016*), with minor modifications. HeLa cells were lysed with the following buffer (4% SDS, 5 mM EDTA, in PBS with complete inhibitor (Roche)). After centrifugation at 100,000 × g for 15 min, supernatant proteins were reduced with 25 mM TCEP for 1 hr at 55°C or at room temperature (RT), and free cysteine residues were alkylated with 20 mM NEM for 3 hr at RT to be blocked. After chloroform/methanol precipitation, resuspended proteins in PBS with 4% SDS and 5 mM EDTA were incubated in buffer (1% SDS, 5 mM EDTA, 1 M NH$_2$OH, pH 7.0) for 1 hr at 37°C to cleave palmitoylation thioester bonds. As a negative control, 1 M Tris-HCl, pH7.0, was used. After precipitation, resuspended proteins in PBS with 4% SDS were PEGylated with 20 mM mPEGs for 1 hr at RT to label newly exposed cysteinyl thiols. As a negative control, 20 mM NEM was used instead of mPEG (-PEG). After precipitation, proteins were resuspended with SDS-sample buffer and boiled at 95°C for 5 min. Protein concentration was measured by BCA protein assay.

### BLUE native PAGE

The PNS of HeLa cells were extracted. The proteins were lysed in 1% Digitonin and passed through a 26 g needle, incubated on ice 30 min, spin 16 000 g for 30 min and run following the manufacturer instructions on the Novex Native PAGE Bis-tris gel system (ThermoFisher).

### Immunofluorescence staining and fluorescence microscope

HeLa cells were seeded on 12 mm glass coverslips (Marienfeld GmbH, Germany) 24 hr prior to transfection. PAT6-myc plasmids were transfected using Fugene (Promega, USA) for 48 hr. Cells were then fixed using 3% paraformaldehyde for 20 min at 37°C, quenched 10 min with 50 mM NH$_4$Cl at RT and permeabilized with 0.1% TX100 for 5 min at RT and finally blocked overnight with 0.5% BSA in PBS. Cells were washed 3x with PBS in between all the steps. Cells were then stained with anti-myc and anti-BiP antibodies for 30 min at RT, washed and incubated again 30 min with their corresponding fluorescent secondary antibodies. Finally cells were mounted on glass slide using mowiol. Imaging was performed using a confocal microscope (LSM710, Zeiss, Germany) with a 63x oil immersion objective (NA 1.4).

### Core model of ZDHHC6 palmitoylation

The ZDHHC6 palmitoylation model (*Figure 5A*) was designed following the approach elaborated previously for calnexin (*Dallavilla et al., 2016*). The core model is based a previously described protein phosphorylation model used in (*Goldbeter and Koshland, 1981*). The set of reaction described by Goldbeter was used to model a single palmitoylation event. Multiple palmitoylation events were modelled replicating this subunit for each reaction.

In the Goldbeter study, the model was mathematically described using mass action terms. Here, due to the presence of multiple modification events, which require the definition of a consistent number of parameters, we described model reactions using so-called 'total quasi-steady state approximation' (tQSSA) (*Pedersen et al., 2008*). With respect to the standard quasi-steady state approximation (QSSA), tQSSA is valid also when the enzyme-substrate concentrations are comparable (*Borghans et al., 1996*). ZDHHC6 is palmitoylated by ZDHHC16. Due to this fact, the use of tQSSA is justified since different proteome studies suggest that DHHC enzymes have similar concentrations (*Beck et al., 2011*; *Merrick et al., 2011*; *Nagaraj et al., 2011*; *Yang et al., 2010*; *Yount et al., 2010*). The step-by-step application of the tQSSA approximation to a palmitoylation model is described in (*Dallavilla et al., 2016*).

The model can be divided in two parts; we first have a phase of synthesis of ZDHHC6, which initially is unfolded (U in *Figure 5A*) and subsequently reaches the folded initially non-palmitoylated ($C^{000}$) form. Each site can subsequently be palmitoylated leading to $C^{100}$ $C^{010}$ and $C^{001}$. These species can then undergo a second palmitoylation step leading to $C^{110}$, $C^{101}$ and $C^{011}$, and a third leading to $C^{111}$. Since palmitoylation is reversible, palmitate can be removed from each of the sites, from all of the species.

ZDHHC6 model is based on the following assumption:

- ZDHHC6 can be degraded in each of its states.
- ZDHHC6 is present in similar concentrations with respect to the modifying enzyme ZDHHC16 (*Beck et al., 2011*; *Merrick et al., 2011*; *Nagaraj et al., 2011*; *Yang et al., 2010*; *Yount et al., 2010*).
- Acyl protein thioesterases (APTs) are more abundant than ZDHHC6 (*Beck et al., 2011*; *Merrick et al., 2011*; *Nagaraj et al., 2011*; *Yang et al., 2010*; *Yount et al., 2010*)
- The three palmitoylation sites may have different affinities with respect to the palmitoylation/depalmitoylation enzymes, therefore we adopted separated Kms and $V_{max}$s for the different sites.
- All the palmitoylation steps are reversible. APT catalyses the depalmitoylation steps.
- Palmitate was considered to be available in excess.

A detailed account of all reactions and differential equations in the model is given in *Supplementary file 1A-C*.

## Parameterization of the model

Since no kinetic data on ZDHHC6 palmitoylation is available, all the parameters of the model were estimated using a genetic algorithm. For the parameter optimization, multiple datasets coming from experimental results were considered. Time course labelling experiments were performed in order to characterize the dynamics of ZDHHC6 synthesis/degradation and incorporation/loss of palmitate.

The types of data used, the description of the Genetic Algorithm (GA) and the step-by-step application of the optimization algorithm used to find values for the parameters are described in detail in (*Dallavilla et al., 2016*). In this paper we use the exact same procedure.

For the parameter estimation we used a calibration set that correspond to the data visible in (*Figure 5B* and *Figure 5—figure supplement 1*). The remaining part of the data (*Figure 5B* and *Figure 5—figure supplement 2*) was used to validate the output of the model and to verify the accuracy of its prediction capabilities.

Because of the presence of multiple objectives there does not exist a single solution that simultaneously optimizes each objective, so the algorithm provides as output a local Pareto set of solutions, which are equally optimal with respect to the fitness function we defined. From the Pareto set provided by the GA and in order to be more accurate and to reduce the variability in the output of the model, we selected the set of parameters that best fitted the calibration data. The procedure for the selection of the subset is described in (*Dallavilla et al., 2016*). In total, 152 different sets of parameter were selected at the end of the optimization.

The results of the optimization can be found in *Supplementary file 1C*.

## Simulating the labelling experiments

Since the accuracy of each parameter set is evaluated by computing the distance between the output of the model and different experiments of the calibration dataset, we made use of a previously a established method to reproduce the different type of experiment in-silico (*Dallavilla et al., 2016*).

## Stochastic simulations

In order to measure the average palmitoylation time of ZDHHC6, we made use of a previously established method to perform in-silico single molecule tracking (*Dallavilla et al., 2016*).

## Conversion of deterministic parameters to stochastic

Parameter conversion from deterministic to stochastic model is needed to perform stochastic simulations (*Dallavilla et al., 2016*). This transformation involves a change of units, from concentration to number of molecules. A single assumption was added to those previously defined (*Dallavilla et al., 2016*), namely that the number of ZDHHC6 molecules per HeLa cell is in the order of 1600 molecules (*Beck et al., 2011*; *Merrick et al., 2011*; *Nagaraj et al., 2011*; *Yang et al., 2010*; *Yount et al., 2010*). *Supplementary file 3A* shows the parameters obtained through the conversion. The model design is shown through the stoichiometry matrix and the propensity function in tables in *Supplementary file 1*.

## Acknowledgements

We thank L Chamberlain and B Martin for sharing their Acyl-RAC protocol, Y and M Fukata for the DHHC plasmids and the PEGylation protocol, P Bastiaens for the APT plasmids and Sylvia Ho for generating the shRNA ZDHHC6 construct. The research leading to these results has received funding from the European Research Council under the European Union's Seventh Framework Programme (FP/2007–2013)/ERC Grant Agreement n. 340260 - PalmERa'. This work was also supported by grants from the Swiss National Science Foundation (to GvdG and to VH), the Swiss National Centre of Competence in Research (NCCR) Chemical Biology (to GvdG) and the Swiss SystemsX.ch initiative evaluated by the Swiss National Science Foundation (LipidX) (to GvdG and to VH). TD is a recipient of an iPhD fellowship from the Swiss SystemsX.ch initiative.

## Additional information

### Funding

| Funder | Grant reference number | Author |
| --- | --- | --- |
| European Research Council | 340260 PalmERa | F Gisou van der Goot |
| Schweizerischer Nationalfonds zur Förderung der Wissenschaftlichen Forschung | SystemsX iPhD Fellowship | Tiziano Dallavilla |
| Schweizerischer Nationalfonds zur Förderung der Wissenschaftlichen Forschung | SystemsX.ch LipidX RTD | Vassily Hatzimanikatis F Gisou van der Goot |
| Schweizerischer Nationalfonds zur Förderung der Wissenschaftlichen Forschung | Division III Grant | F Gisou van der Goot |

The funders had no role in study design, data collection and interpretation, or the decision to submit the work for publication.

### Author contributions

Laurence Abrami, Conceptualization, Formal analysis, Investigation, Methodology, Writing—original draft, Writing—review and editing; Tiziano Dallavilla, Software, Formal analysis, Investigation, Methodology, Writing—original draft, Writing—review and editing; Patrick A Sandoz, Formal analysis, Investigation, Writing—review and editing; Mustafa Demir, Investigation, Writing—review and editing; Béatrice Kunz, Resources, Investigation; Georgios Savoglidis, Formal analysis, Supervision,

Methodology; Vassily Hatzimanikatis, Conceptualization, Software, Supervision, Funding acquisition, Writing—original draft, Writing—review and editing; F Gisou van der Goot, Conceptualization, Formal analysis, Supervision, Funding acquisition, Methodology, Writing—original draft, Project administration, Writing—review and editing

### Author ORCIDs
Patrick A Sandoz (iD) http://orcid.org/0000-0002-8379-7267
Vassily Hatzimanikatis (iD) http://orcid.org/0000-0001-6432-4694
F Gisou van der Goot (iD) http://orcid.org/0000-0002-8522-274X

### Decision letter and Author response
Decision letter https://doi.org/10.7554/eLife.27826.024
Author response https://doi.org/10.7554/eLife.27826.025

---

## Additional files

### Supplementary files
• Source data 1. Source data for all figures.
DOI: https://doi.org/10.7554/eLife.27826.017

• Supplementary file 1. (**A**) Model reactions. The model of DHHC6 palmitoylation contains 35 different reactions, describing synthesis, folding, degradation, and the enzymatic reactions of DHHC6 palmitoylation/depalmitoylation. In the following table we describe in detail how the rates for those reactions are calculated. (**B**) Mass balance equations. The following table describes the mass balance for each of the species of DHHC6 model. The rates of the mass balance of each state are described in detail in **A**. (**C**) Model parameters. The output of GA is a set of optimal solutions, where a solution is a complete set of parameter needed to perform model simulations. From this set we extracted a sub-set of 152 solutions that obtained a GA score better than a set threshold for each objective. During the analysis the model was simulated for each set of parameters of the sub-set. We then reported in this paper the mean of the outputs along with the first and third quartile of their distribution.
DOI: https://doi.org/10.7554/eLife.27826.018

• Supplementary file 2. (**A**) Half-life of DHHC6 in different palmitoylation states. The half-life was estimated from the decay rate constant obtained through parameter estimation. The half-life is calculate as: $\ln(2)/kd$. (**B**) Total amount of protein in steady state relative to WT for WT and CAA mutant in different conditions. The table shows the total protein in steady state in the model relative to the abundance of DHHC6 observed in steady state in WT conditions. Simulations are performed for WT and CAA mutant under control conditions, after overexpression of DHHC16 and after silencing of APT2.
DOI: https://doi.org/10.7554/eLife.27826.019

• Supplementary file 3. (**A**) Parameters used for stochastic simulations. The following parameters were obtained through the conversion of the deterministic parameters estimated by the GA (see *Dallavilla et al., 2016*). (**B**) Stoichiometric matrix used for stochastic simulations. In the following table we define the stoichiometry and the directionality of the reactions of the model. Each reaction has a directionality that define which are the reagents and which are the products. In this table each line represents a model reaction, while in the columns we find all the states of the model. For each reaction the states of the model that take part to the reaction as reagents are marked with $-1$, while the states that participate as products are marked with 1. The matrix that is formed in this way allow to attribute the correct directionality to model reactions during the calculation of the mass balance for each state of the model. (**C**) Propensity function used for stochastic simulations. In the first column of the table, each line describes a reaction of the model. To each reaction is associated a rate, in the second column, that describes the probability of that reaction to happen at each time step of the stochastic simulation.
DOI: https://doi.org/10.7554/eLife.27826.020

• Supplementary file 4. Results of stochastic simulation when APT2 is silenced. The table shows the average number of passages of a DHHC6 molecule in the different palmitoylation states when APT2 is silenced. The time spent in each state is also reported.
DOI: https://doi.org/10.7554/eLife.27826.021

• Transparent reporting form
DOI: https://doi.org/10.7554/eLife.27826.022

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
