## [Decision Letter]

Thank you for submitting your article "Identification and dynamics of the DHHC16-DHHC6 palmitoylation cascade" for consideration by *eLife*. Your article has been reviewed by three peer reviewers, one of whom, Philippe IH Bastiaens (Reviewer #1), is a member of our Board of Reviewing Editors, and the evaluation has been overseen by Ivan Dikic as the Senior Editor. The following individual involved in review of your submission has agreed to reveal their identity: Anthony Magee (Reviewer #3).

The reviewers have discussed the reviews with one another and the Reviewing Editor has drafted this decision to help you prepare a revised submission.

Summary:

This very interesting multidisciplinary study deals with a palmitoylation/depalmitopyaltion reaction network around the palmitoyl transferase DHHC6 that controls its DHHC16 induced PAT activity. The work presents a very thorough study that led to the identification of the first palmitoylation cascade. This is a very exciting discovery, which poses many interesting questions at multiple levels, from basic biochemistry and cell biology, to the systems level properties of palmitoylation cascades.

Essential revisions:

1) Are the different palmitoylated species of DHHC6 differentially localized and thereby palmitoylate different substrates? The authors should analyse the localisation of the different palmitoylation deficient mutants of DHHC6 and possibly investigate their activity on one of the other substrates of DHHC6 to clarify that the system is not a substrate selection network.

2) It is unclear why the authors had to use stochastic simulation for part of their analyses given the copy numbers of the analyzed molecules. Can the authors use their model to propose experiments that would make the model parameters more identifiable?

3) The authors do not measure enzyme activity directly; rather it is assessed indirectly by in vivo labelling of substrates. This should be discussed and addressed. Ideally an in vitro direct assay of DHHC6 activity should be used.

4) The methodology leave open the possibility that these proteins do not genuinely interact physically but could simply be associated with the same detergent-insoluble structures. More stringent solubilisation methods (e.g. stronger detergents or the extraction of cholesterol with methyl-β-cyclodextrin) should be used to confirm the result.

5) The pulse is relatively long as compared to the chase in some pulse-chase experiments (such as Figure 2 a 2h pulse is used). This can incorrectly estimate the rate of turnover. This point should be addressed and discussed.

---

## [Author Response]

*Essential revisions:*
*1) Are the different palmitoylated species of DHHC6 differentially localized and thereby palmitoylate different substrates? The authors should analyse the localisation of the different palmitoylation deficient mutants of DHHC6 and possibly investigate their activity on one of the other substrates of DHHC6 to clarify that the system is not a substrate selection network.*

That the different palmitoylation species of DHHC6 may have different substrate specificities is an interesting point that we now mention, but rule out (see below), in the manuscript.

Localization: We have analyzed the localization of WT DHHC6 and the various mutants and find that all stainings are very similar (new figure: Figure 4—figure supplement 1). This distribution might represent the situation of ectopic expression, but this is also situation in which the activity of the mutants was measured. Thus the differences in target protein modification do not correlate with differences in localization.

Additional substrates: We have now monitored the palmitoylation of another endogenous DHHC6 target, the transferrin receptor, by the various DHHC6 cysteine mutants. We found that the differential activity of the mutants is the same as for calnexin (new figures: Figure 4), further supporting that the palmitoylated species have different activities rather than different targets.

*2) It is unclear why the authors had to use stochastic simulation for part of their analyses given the copy numbers of the analyzed molecules.*

We used stochastic simulations because we wanted to quantify how much time each protein molecule spends in the different states and the transition dynamics between the different states. While the deterministic simulation could provide an estimate of these properties, the stochastic simulations in addition provides a quantification of the distribution of the times of interest and allowed us to ask questions about “individual” molecules. Moreover, the concentration of the species is in the range of nanomolar concentration, which is the range where the intrinsic noise is manifested and it is also observed in our simulations. This is now mentioned in the text.

*Can the authors use their model to propose experiments that would make the model parameters more identifiable?*

A number of experiments predicted by the earlier versions of the model are actually already part of the paper. For example, initial protein decay experiments were performed with a classical (for our lab) 20 min metabolic 35S Cys/Met pulse. Under these conditions we could not detect any effect of palmitoylation on apparent half-lives when comparing WT DHHC6 and the various cysteine mutants. The model then predicted that palmitoylation on the first cysteine leads to major destabilization, for which we had no experimental evidence at that point, and that longer 35S Cys/Met pulse periods would reveal differences in apparent decay kinetics between the WT protein and the various cysteine mutants. These predictions turned out to be correct and subsequently led us to test overexpression of DHHC16 and silencing of APT2 on decay kinetics. This is now mentioned in the text.

*3) The authors do not measure enzyme activity directly; rather it is assessed indirectly by* in vivo *labelling of substrates. This should be discussed and addressed. Ideally an* in vitro *direct assay of DHHC6 activity should be used.*

Having an in vitro assay for DHHC6 activity would indeed show that calnexin or the transferrin receptor are directly modified by DHHC6. Unfortunately the fact that DHHC6 is a membrane protein, that it requires palmitoylation by another membrane protein DHHC16 activity for and the fact that all identified DHHC6 substrates are so far membrane proteins make the establishment of an in vitro assay extremely challenging and beyond the scope of this study. We have now modified the text throughout the manuscript to avoid using the word “activity” incorrectly and rather refer to the ability of DHHC6 to mediated modification of reported targets, such as calnexin and transferrin receptor, in cells.

*4) The methodology leave open the possibility that these proteins do not genuinely interact physically but could simply be associated with the same detergent-insoluble structures. More stringent solubilisation methods (e.g. stronger detergents or the extraction of cholesterol with methyl-β-cyclodextrin) should be used to confirm the result.*

It is not clear to us to what experiment the reviewers are referring. We assume it is either the co-IP of WT and AAA mutant or the blue native gels.

We have now carefully reworded the text to include the two options, i.e. that WT and AAA mutant either form different complexes or reside in different membrane domains.

*5) The pulse is relatively long as compared to the chase in some pulse-chase experiments (such as Figure 2 a 2h pulse is used). This can incorrectly estimate the rate of turnover. This point should be addressed and discussed.*

Indeed the pulse time has a major influence on the apparent turnover rate, both for 35S Cys/Met and 3H-palmitate labeling. We have now clearly conveyed this point in the text. For ^35^S pulse-chase experiments, we have actually performed experiments changing the pulse time in order to calibrate and validate the model (see point 2 above). We have specified that the protein turn over rates that we provide in Figure 5 correspond to the turnover rates calculated by the model, which are independent of labeling time, not apparent turnover rates determined experimentally and which are indeed heavily influenced by labeling time.